# Hydrogel dressings with intrinsic antibiofilm and antioxidative dual functionalities accelerate infected diabetic wound healing

Dicky Pranantyo [1,2], Chun Kiat Yeo[1,3], Yang Wu [1], Chen Fan[4,5], Xiaofei Xu[1], Yun Sheng Yip[6], Marcus Ivan Gerard Vos[6], Surendra H. Mahadevegowda [1], Priscilla Lay Keng Lim [4], Liang Yang [7], Paula T. Hammond [2,8], David Ian Leavesley [4], Nguan Soon Tan [6,9] ✉ & Mary B. Chan-Park [1,9] ✉

Chronic wounds are often infected with biofilm bacteria and characterized by high oxidative stress. Current dressings that promote chronic wound healing either require additional processes such as photothermal irradiation or leave behind gross amounts of undesirable residues. We report a dual-functionality hydrogel dressing with intrinsic antibiofilm and antioxidative properties that are synergistic and low-leaching. The hydrogel is a crosslinked network with tethered antibacterial cationic polyimidazolium and antioxidative *N*-acetylcysteine. In a murine diabetic wound model, the hydrogel accelerates the closure of wounds infected with methicillin-resistant *Staphylococcus aureus* or carbapenem-resistant *Pseudomonas aeruginosa* biofilm. Furthermore, a three-dimensional ex vivo human skin equivalent model shows that *N*-acetylcysteine promotes the keratinocyte differentiation. Our hydrogel dressing can be made into different formats for the healing of both flat and deep infected chronic wounds without contamination of the wound or needing other modalities such as photothermal irradiation.

Normal wounds progress through four main stages of healing: coagulation, inflammation, proliferation, and maturation[1–3]. Chronic wounds, on the other hand, are often trapped in the inflammatory stage so that they fail to progress towards healing[4]. Chronic wounds include various ulcer types, including diabetic foot ulcers, venous leg ulcer wounds and pressure ulcers[5]. The economic cost of all chronic non-healing wounds in the US alone is estimated to be more than $50 billion per year. Diabetic foot ulcers have a 5-year mortality rate (30.5%) comparable to that of cancer (31%)[6,7]. As populations, especially in developed countries, grow and age, the prevalence and impact of chronic wounds is expected to further increase.

Conventional wound dressings are not designed to promote the closure of hard-to-heal chronic wounds. A common characteristic of chronic wounds is prolonged inflammation[8,9]. Various studies indicate that non-healing wounds are trapped in a chronic inflammatory state that inhibits the normal progress of healing[4]. Specifically, recent investigations of chronic wound tissues and fluids indicate competition between pro-inflammatory and anti-inflammatory signals that

[1]Centre for Antimicrobial Bioengineering, School of Chemistry, Chemical Engineering and Biotechnology, Nanyang Technological University, 62 Nanyang Drive, Singapore 637459, Singapore. [2]Antimicrobial Resistance Interdisciplinary Research Group, Singapore-MIT Alliance for Research and Technology, Singapore 138602, Singapore. [3]NTU Institute for Health Technologies, Interdisciplinary Graduate School, Nanyang Technological University, Singapore 637553, Singapore. [4]Skin Research Institute of Singapore, Agency for Science, Technology and Research (A*STAR), 11 Mandalay Road, Singapore 308232, Singapore. [5]Wenzhou Institute, University of Chinese Academy of Sciences, Wenzhou, Zhejiang 325000, China. [6]Lee Kong Chian School of Medicine, Nanyang Technological University, 59 Nanyang Drive, Singapore 636921, Singapore. [7]School of Medicine, Southern University of Science and Technology, Shenzhen 518055, China. [8]Department of Chemical Engineering, Massachusetts Institute of Technology, Cambridge, MA 02142, USA. [9]School of Biological Sciences, Nanyang Technological University, 60 Nanyang Drive, Singapore 637551, Singapore. ✉e-mail: nstan@ntu.edu.sg; mbechan@ntu.edu.sg

leads to a redox imbalance and prevention of the proper wound healing occurrence[10,11]. This locks the wound into a state of chronic inflammation that hinders progression to wound closure. Due to the persistent inflammation, infiltrating neutrophils and macrophages dwell in the site and they secrete reactive oxygen species (ROS) to combat colonization by microorganisms. The elevated levels of ROS in chronic wounds have negative effects on wound healing as they may cause damage to cells, tissues, and the extracellular matrix (ECM), and activate latent extracellular proteases (such as matrix metalloproteinases (MMPs)) and inflammatory cytokines[12]. The prolonged state of elevated inflammation and levels of ROS cause the wound to be unable to escape the inflammatory phase. In severe cases, cells within and adjacent to the wound bed undergo programmed cell death (*i.e.* apoptosis, pyroptosis, ferroptosis) due to the high oxidative stress, triggering a cascade of events in neighboring cells leading to the same fate. This is likely the reason many chronic wounds become necrotic, which necessitates drastic procedures such as tissue debridement or worse, amputation, to protect the patient's life.

To overcome the high oxidative stress of chronic wounds, many researchers have explored the application of topical antioxidants[13,14], such as formulations using curcumin and *N*-acetyl-L-cysteine (NAC)[15–17]. However, it was noted that the most common challenge in chronic wound care was not addressed in such wound rinse/dressing designs, which is that many wounds are colonized with biofilm-forming bacteria. Biofilm infections have been found to delay healing and reduce the clinical effectiveness of topical antioxidant solution[17,18]. Biofilms, rather than planktonic bacteria, are the main form of microorganisms colonizing chronic wounds[19] and they are often recalcitrant to treatment[20]. Biofilm infections in wounds are most commonly caused by methicillin-resistant *Staphylococcus aureus* (MRSA) and *Pseudomonas aeruginosa* (*P. aeruginosa*)[21]. Biofilms have the capability to absorb nutrients from the ECM, such as carbon, nitrogen and phosphate, to supply their growth[22]. Mature biofilms form extracellular polymeric substances (EPS) that surround the bacteria and protect them from countermeasures such as host immune cells, macrophages, antimicrobial peptides and antibiotics[23]. These countermeasures fail to attack the biofilm bacteria because they cannot penetrate the protective EPS layer in sufficient quantity to have therapeutic effect. Biofilms are often stable so that they remain on wound surfaces for as long as they are undisturbed by medical interventions.

Advanced chronic wound dressings should be designed for comprehensive treatments, addressing both antioxidative and antibacterial/antibiofilm requirements. Although previous studies have explored possible solutions, each approach has encountered limitations. Hitherto, no dual-functionality wound treatments is standalone or non-contaminating because they rely on either the innate immune response, photothermal irradiation, antibiotic release or metal oxides[24–27]. Our wound dressing utilizes the porous nature of the hydrogel, combined with the intrinsic antibacterial and antioxidative properties of cationic polymers and NAC, respectively. It is standalone and features ultra-low leaching, making it both convenient and safe to use.

Shiekh et al. developed an exosome-laden, oxygen-releasing cryogel, OxOBand, which demonstrated enhanced collagen deposition, re-epithelialization, neo-vascularization, and reduced oxidative stress in diabetic wounds[24]. However, it relied on oxygen release to stimulate production of ROS (such as $H_2O_2$ and $O^{2-}$) by the host macrophages to combat bacterial infections, which may be dampened in immunocompromised patients. It also needed adipose-derived stem cells for the exosome creation which are associated with large-scale manufacturing limitations and various ethical, regulatory and safety issues. Another approach involved a nanocomposite containing molybdenum disulfide ($MoS_2$) nanosheets and cerium dioxide ($CeO_2$) nanoparticles (NPs) to deliver both the photothermal antibacterial effect and the antioxidative function for treating infected wounds[25].

This topical ointment required 808-nm laser treatment to activate the antibacterial effect of $MoS_2$ and left metal components (Mo and Ce) on the host body. Another composite hydrogel, comprising antioxidative (poly)dopamine-modified gelatin, carbon nanotubes (CNTs), and antibiotic doxycycline, also showed promise for effecting antibacterial and antioxidant properties[26]. However, the photothermal antibacterial effect of the CNTs relied on NIR irradiation, and the release of the antibiotic from the dressing raised concerns about the potential drug resistance development. A thermoresponsive hydrogel based on a triblock copolymer of caprolactone, glycolide, and ethylene glycol with polydopamine and silver NPs modifications exhibited efficacy against Gram-positive *S. aureus* but not against Gram-negative *P. aeruginosa* bacteria[27] Ideal chronic wound dressings would exhibit both anti-inflammatory and broad-spectrum anti-infection functions without necessitating additional interventions such as irradiation and would not contaminate the wound site. Current reported treatments do not meet all these requirements.

We report herein a dual-functionality synthetic hydrogel, PPN, with intrinsic antibiofilm and antioxidant properties that synergistically act to support healing of infected diabetic wounds, as demonstrated with murine models. The standalone PPN synthetic hydrogel, which does not require any additional modality or further process, comprises a crosslinked polyethylene glycol (**P**EG) hydrogel tethered with highly potent antibacterial cationic polymer, polyimidazolium (**P**IM), and the antioxidant *N*-acetylcysteine (**N**AC). This robust hydrogel is ultralow-leachable and completely devoid from antibiotics, metal compounds, or nanoparticles, ensuring minimal residue at the wound site after dressing removal. Studies on the PPN in a 3D human skin equivalent model shows that NAC promotes re-epithelialization and keratinocyte differentiation, while the incorporation of the PIM derivative does not hinder the wound healing process.

## Results

A polyethylene glycol (PEG)-derived hydrogel was supplemented with antibiofilm and antioxidant properties through the addition of a cationic main chain polyimidazolium-maleimide (PIM-Mal) and a second component of *N*-acetylcysteine (NAC) (Fig. 1a). Both molecules (PIM-Mal derivative and NAC) were covalently linked into the hydrogel network through the thiol-maleimide chemistry. PIM-Mal was pre-synthesized via the facile Poly-Radziszewski chemistry to make the diamine-terminated PIM followed by end modification with maleic anhydride[28,29] (Supplementary Fig. 1). The hydrogel was made into two formats, specifically the film and fiber formats. The first format, a film hydrogel format labeled as PPN (Fig. 1b) which was obtained from crosslinking a 4-arm PEG-thiol (PEG-4SH) with a 4-arm PEG-maleimide (PEG-4Mal), together with the addition of PIM-Mal and NAC, is suitable for relatively flat wounds. The second format utilizing alginate (labeled as Alg-PPN (Fig. 1c)) was assembled by crosslinking (i) PEG-2Mal-2NAC made from PEG-4Mal that was pre-reacted with NAC so that two arms were attached with NAC (Supplementary Fig. 2), and (ii) Alg-SH-PIM that was pre-reacted from thiol-functionalized alginate (Alg-SH) with PIM-Mal; this format is suitable for deep wounds.

Photographs of the two formats, i.e., film and alginate (Alg) fiber, are shown in Fig. 2a, b. We performed initial testing with the hydrogel film format (Figs. 1b, 2a). The cationic PIM(C4)-Mal, which has a butyl (C4) linker between the imidazolium rings, was pre-synthesized (Supplementary Fig. 1). The $^1$H NMR characterization of PIM(C4)-Mal is presented in Supplementary Fig. 3a. The molecular weight ($M_n$) of PIM(C4)-Mal as measured by gel permeation chromatography was 2766 Da (Supplementary Fig. 4). The minimum inhibitory concentration (MIC) of PIM(C4)-Mal varies from 2 to 8 $\mu$g/mL (Supplementary Table 1) when tested against various ESKAPE bacteria (specifically *E. faecium* 19434, methicillin-resistant *S. aureus* (MRSA BAA-40 and

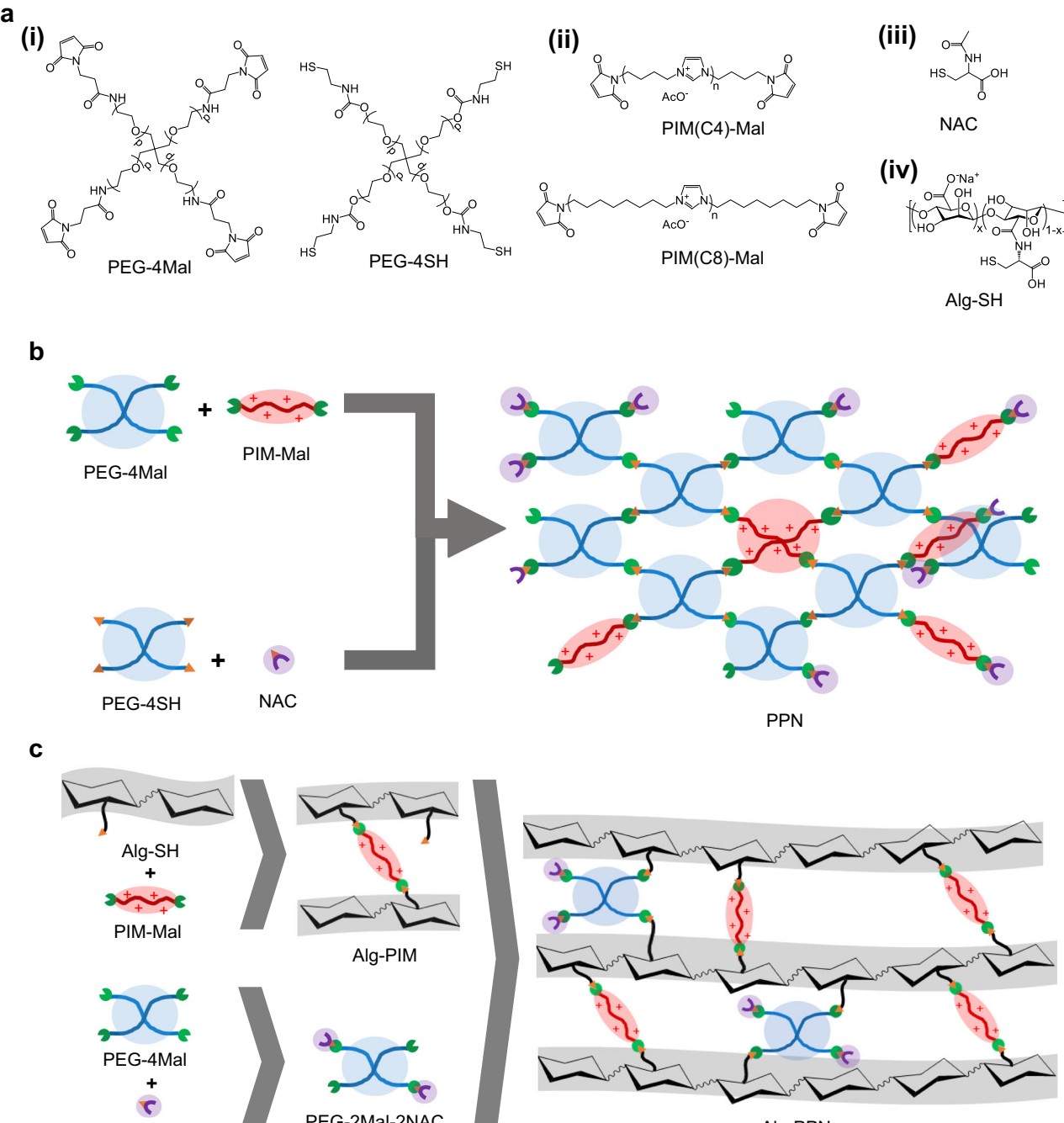

**Fig. 1 | Preparation schemes of the film and fiber hydrogel wound dressings. a** Structures of the main components in hydrogel film and fiber dressings. Syntheses of (**b**) PPN hydrogel film and (**c**) Alg-PPN hydrogel fiber dressings.

MRSA USA300), *E. cloacae* 13047, *K. pneumoniae* 13883, *P. aeruginosa* 01 (PAO1), and carbapenem-resistant *P. aeruginosa* and *A. baumannii* (CR-PA and CR-AB)), indicating that the polymer is a potent growth inhibitor for a wide spectrum of Gram-positive and Gram-negative bacteria.

We tested various compositions (Table 1) of the film hydrogels. The dual-functional film hydrogels (i.e., containing both PIM(C4)-Mal and NAC) are labeled as PEG-PIM-NAC (PPN(C4)-x), where the suffix (x) refers to the PIM(C4)-Mal concentration (x mg/mL) (Table 1). A control hydrogel (denoted as PPcontrol) was assembled by mixing PEG-4SH with PEG-4Mal in deionized (DI) water without the active components PIM and NAC. Single-functional hydrogels excluding either PIM (labeled as PP-N) or NAC (PPN-) were also prepared. The film hydrogel

network was formed mainly through the reaction of PEG-4SH with PEG-4Mal; NAC and a portion of the PIM(C4)-Mal were tethered to the network as pendant molecules (Fig. 1b). The hydrogel precursors were able to crosslink in DI water at 25 °C in less than one minute[30,31]. The hydrogel films showed a tensile strength ranging from 4 to 5 kPa and tensile strain (elongation) between 50 and 58% (Fig. 2c(i), d(i)), which is slightly lower than human skin stretchability (60–75%). In an aqueous environment, the film hydrogels swelled, absorbing 8–10 times their initial mass of water within 20 min; thereafter the swelling kinetics plateaued to reach 10–12 times within 60 min (Fig. 2e, f, Supplementary Fig. 5a, b). The swollen hydrogels were stable and exhibited constant mass after incubation for 7 days in bacterial extracts (Supplementary Fig. 6a-b) and infected wound fluids (Supplementary

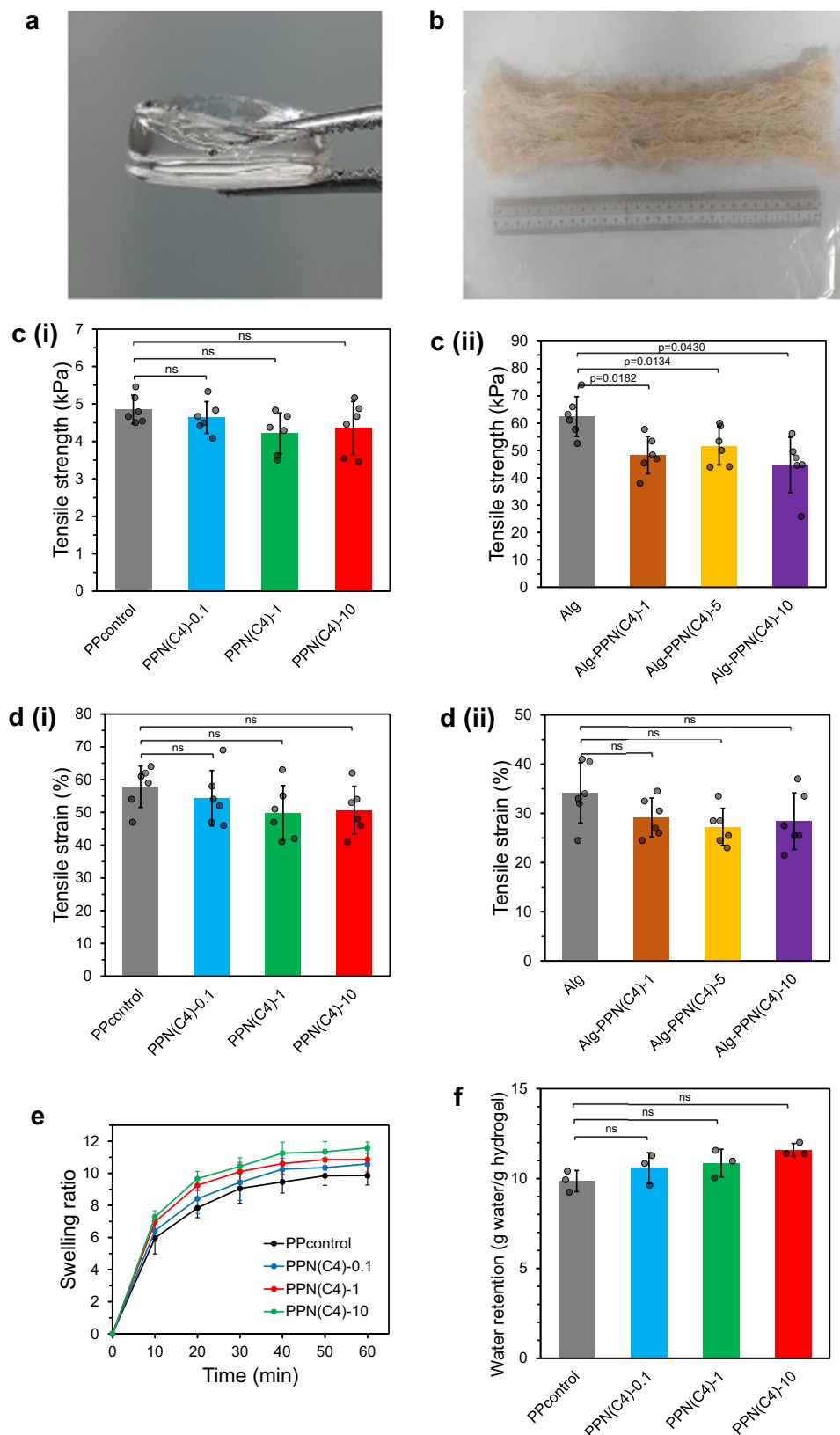

**Fig. 2 | Physical characterizations of the film and fiber hydrogel dressings.** Visual appearance of the wound dressings investigated: (**a**) PPN(C4)−1 film hydrogel and (**b**) Alg-PPN(C8)-5 alginate fiber hydrogel. Tensile strength of the (**c**(i)) film hydrogels and (**c**(ii)) alginate fiber hydrogels ($n = 6$ independent experiments, two-tailed Student's $t$ test). Tensile strain (elongation) of the (**d**(i)) film hydrogels and (**d**(ii)) alginate fiber hydrogels ($n = 6$ independent experiments, two-tailed Student's $t$ test). **e** Swelling kinetics (mass increase/initial mass versus time) of the film hydrogels ($n = 3$ independent experiments). **f** Water retention capacity of the film hydrogels after 1 day immersion in water ($n = 3$ independent experiments, two-tailed Student's $t$ test). Data are presented as mean values ± SD.

**Table 1 | Compositions of the PPN(C4) film hydrogels in 1 mL of DI water**

| Hydrogel formulation | PEG-4SH | PEG-4Mal | PIM(C4)-Mal | NAC |
|---|---|---|---|---|
| PPcontrol | 50 mg (2.5 μmol) | 50 mg (2.5 μmol) | – | – |
| PPN(C4)–0.1 | 50 mg (2.5 μmol) | 50 mg (2.5 μmol) | 0.1 mg (0.04 μmol) | 0.2 mg (1 μmol) |
| PPN(C4)-1 | 50 mg (2.5 μmol) | 50 mg (2.5 μmol) | 1 mg (0.36 μmol) | 0.2 mg (1 μmol) |
| PPN(C4)-10 | 50 mg (2.5 μmol) | 50 mg (2.5 μmol) | 10 mg (3.62 μmol) | 0.2 mg (1 μmol) |
| PP-N | 50 mg (2.5 μmol) | 50 mg (2.5 μmol) | – | 0.2 mg (1 μmol) |
| PPN- | 50 mg (2.5 μmol) | 50 mg (2.5 μmol) | 1 mg (0.36 μmol) | – |

Fig. 6c). In the in vivo (murine model) test for infected wounds, the initially transparent hydrogels turned yellowish brown after 2 days, presumably due to absorption of wound fluid and dead bacteria, but remained intact and stable (Supplementary Fig. 7). This proved that the hydrogels are resistant to degradation by infected wound fluids.

In vitro contact killing efficacies of the PPN(C4) hydrogels were measured for various multidrug resistant (MDR) Gram-negative and Gram-positive bacterial strains that are clinically relevant to infected wounds[21], specifically MRSA USA300, CR-AB, PAO1, and CR-PA, which are pathogens of great concern worldwide[12,32]. PPN(C4)–1 and PPN(C4)-10 completely eradicated the various bacterial strains loaded onto the hydrogel discs in 1 h (Fig. 3a–d). PPN(C4)-0.1 hydrogel did not completely eradicate all bacteria (Fig. 3a–d), probably due to its lower concentration of the bioactive antibacterial PIM-Mal component. PPcontrol did not exhibit bactericidal properties. The hydrogels and their extracts also exhibit low toxicity to human dermal fibroblasts (HDFs), as assessed via 3-(4,5-dimethylthiazol-2-yl)-2,5-diphenyltetrazolium bromide (MTT) testing. Cell viability was 100% for HDF exposed to all the hydrogel extracts (Fig. 3e). For the hydrogel contact assay, the viabilities of HDF were 97%, 94% and 89% for PPN(C4)-0.1, PPN(C4)-1 and PPN(C4)-10, respectively (Fig. 3e), indicating low acute toxicity and good biocompatibility of these hydrogels.

PPN(C4)-1 was chosen for further characterization as it exhibited antibacterial efficacy at a lower concentration of PIM(C4)-Mal. We further investigated the effects of the individual bioactive component of PIM or NAC, as well as their combined effects. We formulated PPN(C4)-1-derived hydrogels lacking either PIM or NAC: PP-N was synthesized without PIM(C4)-1, while PPN- was synthesized without NAC (Table 1). Using an ex vivo reconstructed 3D tissue model, the effects of these PPN(C4)-1-derived hydrogels on wound closure, re-epithelialization, and proliferation and differentiation of human keratinocytes in the wound healing process were studied over 7 days. Specifically, we utilized the 3D De-Epidermised Dermis Human Skin Equivalent (DED-HSE) model, a living ex vivo tissue construct in which decellularized dermal scaffolds from human donors are repopulated with allogeneic donor keratinocytes. In addition to being physiologically similar to in vivo skin tissues, as evidenced by Xie et al.[33], the DED-HSE construct also supports the high proliferation and differentiation potential of allogeneic donor keratinocytes owing to the presence of an intact basement membrane[34].

Using the mitochondrial reduction of MTT applied to the 3D DED-HSE model, the effects of hydrogels on the overall healing of excisional wounds were assayed (Fig. 4a, b, Supplementary Fig. 8). The initial wound size (at day 0) is denoted as 100% in Fig. 4b. No wound closure was evident with the silver dressing, as the wound area remained large (~100%). Some wound closures were observed in the untreated control and all hydrogel treatment groups at both 4 and 7 days after wounding. For example, with our various hydrogel formulations at day 7, the remaining unhealed areas were approximately 56–61% of the size of the initial wound. No statistical difference of wound sizes was found between the various hydrogel treatment groups and the PPcontrol (PEG gel without both NAC and PIM) at both day 4 and day 7 (Fig. 4b). These data from the DED-HSE model suggest that the hydrogel

formulations (PPcontrol, PP-N, PPN- and PPN(C4)-1) possess excellent biocompatibility compared to the current commercial silver dressing.

We examined the gross anatomy of DED-HSE samples exposed to the hydrogel formulations (PPcontrol, PP-N, PPN- and PPN(C4)-1). The basic architecture revealed by hematoxylin and eosin (H&E) staining on day 7 identifies a growing wedged-shaped epithelial tongue and greater volume of stratified epithelium in all treatment groups, with the exception of the silver dressing sample. The DED-HSE samples treated with PPN- and PPN(C4)-1, both of which contain PIM(C4)-Mal, exhibited statistically similar epidermal thicknesses compared with the PPcontrol sample (Fig. 4c, Supplementary Table 2), indicating that the incorporation of PIM(C4) did not retard re-epithelization. Notably, wounds treated with PP-N exhibited a 35% thicker epidermis than PPcontrol-treated wounds (Supplementary Table 2); this may be attributable to the healing promotion effect of the NAC component. These data confirm that PP-N, PPN- and PPN(C4)-1 are biocompatible, do not attenuate the re-epithelialization process, and that PP-N with NAC incorporation accelerates wound re-epithelialization.

The proliferation of cells in DED-HSE samples exposed to the different hydrogel formulations was also corroborated by the presence of the nuclear transcription factor p63, expressed by undifferentiated proliferating keratinocytes. The brown dots evident in Fig. 4c (second row) are p63-expressing cells, indicating that these individual cells are undergoing proliferation, while the blue dots indicate the nuclei of both proliferative and non-proliferative cells. On day 7, intense immunoreactivity was evident in the keratinocyte populations within the basal layers of DED-HSEs from all treatment groups (Fig. 4c, second row). DED-HSE samples exposed to PP-N, PPN-, and PPN(C4)-1 exhibited statistically comparable numbers of p63-positive cells as compared with DED-HSE exposed to PPcontrol (Supplementary Table 2). This finding suggests that the hydrogels incorporating the individual or combined components of PIM(C4) or NAC did not affect keratinocyte proliferation. Taken together, the presence of NAC or PIM does not retard proliferation, as shown in the H&E and p63 assays, and the MTT assay shows that the combination does not have any negative effect on proliferation, which is in contrast to silver dressing.

To evaluate the potential impact of PP-N, PPN- and PPN(C4)-1 on the maturation of keratinocytes, we also examined the expression of two epithelial-specific markers. The cytokeratin 10 (K10) protein is present in all suprabasal cell layers, including the stratum corneum; it is exclusively expressed by keratinocytes undergoing squamous differentiation. The cytokeratin 14 (K14) protein is exclusively expressed by the basal undifferentiated keratinocytes. The relative distributions of K10 and K14 illustrate the life-cycle of keratinocytes as they transition from proliferating to non-proliferating, differentiated states. We found that DED-HSE treated with PPN- expressed comparable K10 and K14 signal intensities ($p > 0.05$) to DED-HSE treated with PPcontrol (Fig. 4c, Supplementary Table 2). Interestingly, DED-HSE treated with PP-N exhibited a 53% stronger signal intensity for K14 compared to PPcontrol ($p \leq 0.05$), while showing a 138% increase in signal intensity for K10 ($p \leq 0.05$, Supplementary Table 2), suggesting that PP-N, enriched with NAC, predominantly fosters keratinocyte proliferation. Additionally, DED-HSE treated with PPN(C4)-1 displayed a 109% increase in signal intensity for K10 compared to PPcontrol ($p \leq 0.05$),

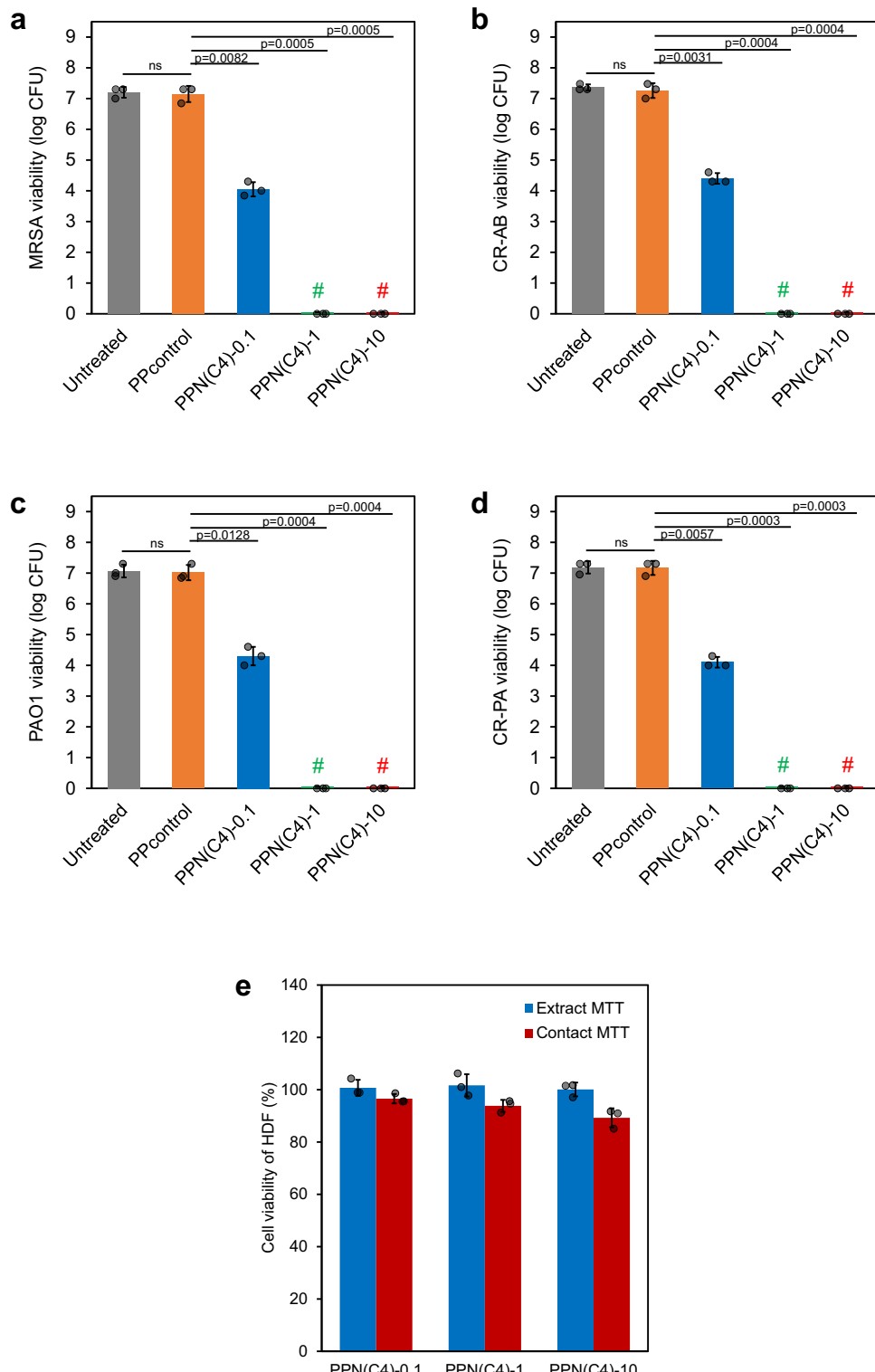

**Fig. 3 | Film hydrogels exhibited in vitro antibacterial activity and cytocompatibility. a–d** Antibacterial efficacy of PPN(C4) film hydrogels assessed via contact testing for 1 h. # denotes that no bacterial colonies were observed on the agar plate (*n* = 3 biologically independent samples, two-tailed Student's *t* test). **e** Cell viability of human dermal fibroblasts (HDFs) after incubation with the extracts (blue) or direct contact immersion (red) of PPN(C4) film hydrogels at 37 °C for 24 h (*n* = 3 cells examined over 3 independent experiments). Data are presented as mean values ± SD.

with a comparable (*p* > 0.05) signal intensity observed for K14. This finding indicates that PP-N (containing NAC) promotes keratinocyte proliferation, which may correlate with better overall re-epithelialization as observed by H&E staining. These data suggest that hydrogels incorporating PIM(C4) and NAC are biocompatible, do

not hinder the differentiation process, and that NAC in the composite PPN(C4)-1 might enhance the squamous differentiation of keratinocytes.

Subsequently, we assessed the hydrogels in vivo using a diabetic murine model of infected wounds to evaluate their efficacy in clearing

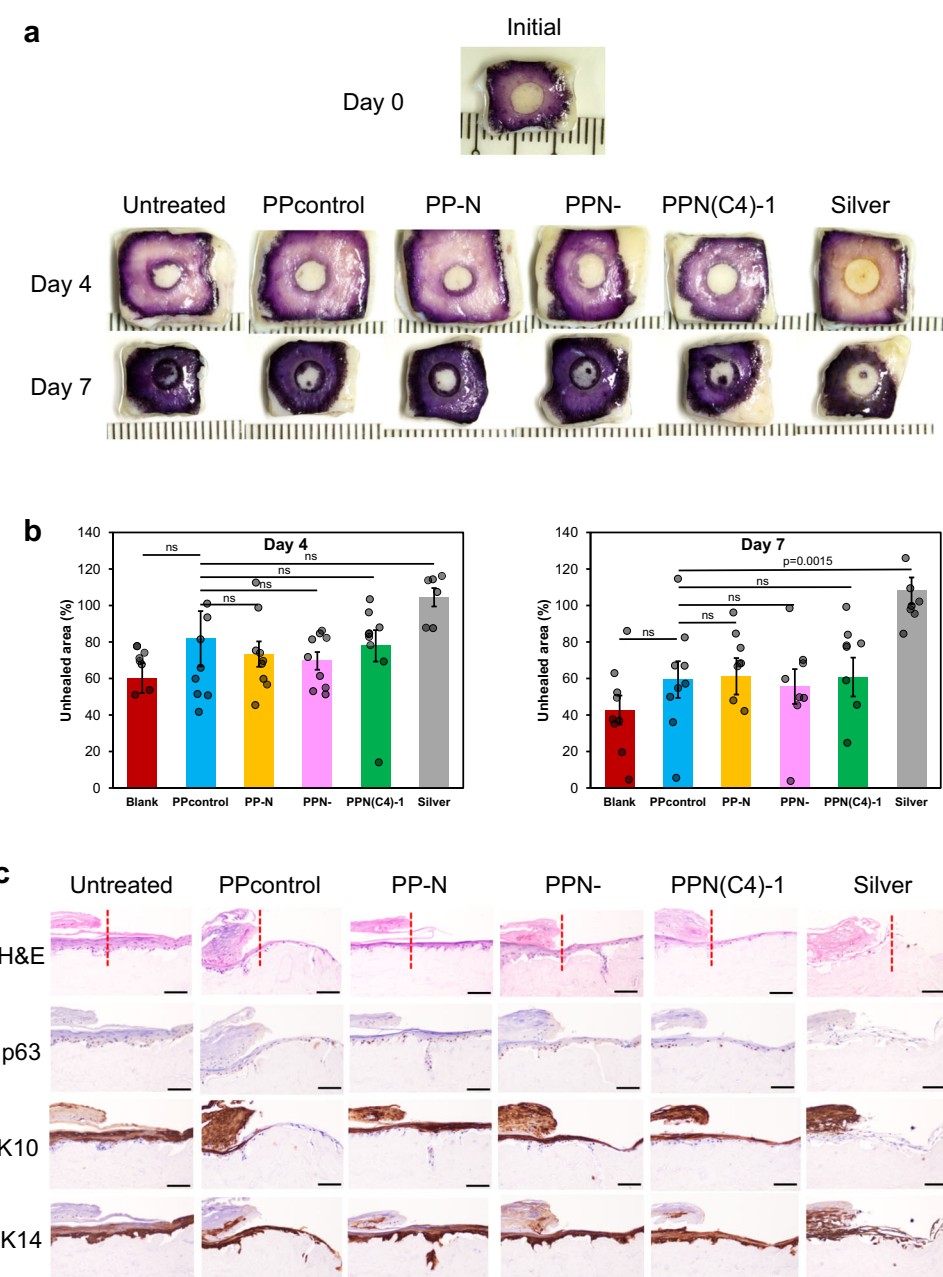

**Fig. 4 | Ex vivo human skin equivalent (HSE) model of the film hydrogels.**
**a** Representative images of MTT staining for wound healing. **b** Quantitative mea-
surement of the MTT assay at the day 4 and day 7 time points (triplicated; $n = 6–9$
biologically independent samples, two-tailed Student's $t$ test, data are presented as
mean values ± SEM). **c** Representative immunostaining images of H&E, p63, Keratin

10 (K10), and Keratin 14 (K14) after treatment with the PPN(C4)−1 hydrogel and its
variants (PP-N is without PIM(C4)−1, and PPN- is without NAC). The samples were
seriallly sectioned from the same embedded sample and each sample slice with
section thickness = 4 µm was used only for one type of staining. Scale bar = 100 µm.

biofilm bacteria, employing various MDR strains, namely MRSA USA-
300, PAO1, CR-AB, and CR-PA, with a commercial silver-based anti-
microbial wound dressing serving as a control. Wounds treated with
the PPN(C4)-1 hydrogel recorded greater than 3 log reduction (>99.9%)
in the colonization of all bacterial strains tested (Fig. 5a). This perfor-
mance surpasses the generally suboptimal treatments (0.1–0.3 log
reduction) observed with the silver dressing and PPcontrol (Fig. 5a). To
conduct a more detailed analysis of the biofilm within the wounds, we
initiated an in vivo wound experiment using fluorescent-labeled bac-
teria, specifically the MRSA USA300 strain (AH1263) carrying the
pHC47 plasmid that expresses red fluorescent mCherry gene[35]. After a
24-h treatment period, we examined the wound lesions using confocal
microscopy. The results revealed fluorescent biofilms with a thickness

of up to 40 µm in the wounds (Supplementary Fig. 9a–d). Based on the
bacterial count data (Fig. 5b) and biofilm volume, the estimated bac-
terial density in the untreated biofilm was 0.1 CFU/µm³. Among all
sample groups, wounds treated with the PPN(C4)-1 hydrogel showed
the lowest number of residual fluorescent bacteria in the biofilms,
corroborating that PPN(C4)-1 treatment decreased the density of
biofilm bacteria.

We also studied the dynamics of biofilm bacteria (MRSA USA300)
count and wound size reduction at the murine diabetic wound site
over a 2-week period of treatment with PPN(C4)-1 and controls. The
wounds depicted in Fig. 5b were collected similarly to those in Fig. 5a,
with mice being sacrificed and the full wounds excised and homo-
genized each day for bacterial quantification. The largest reduction of

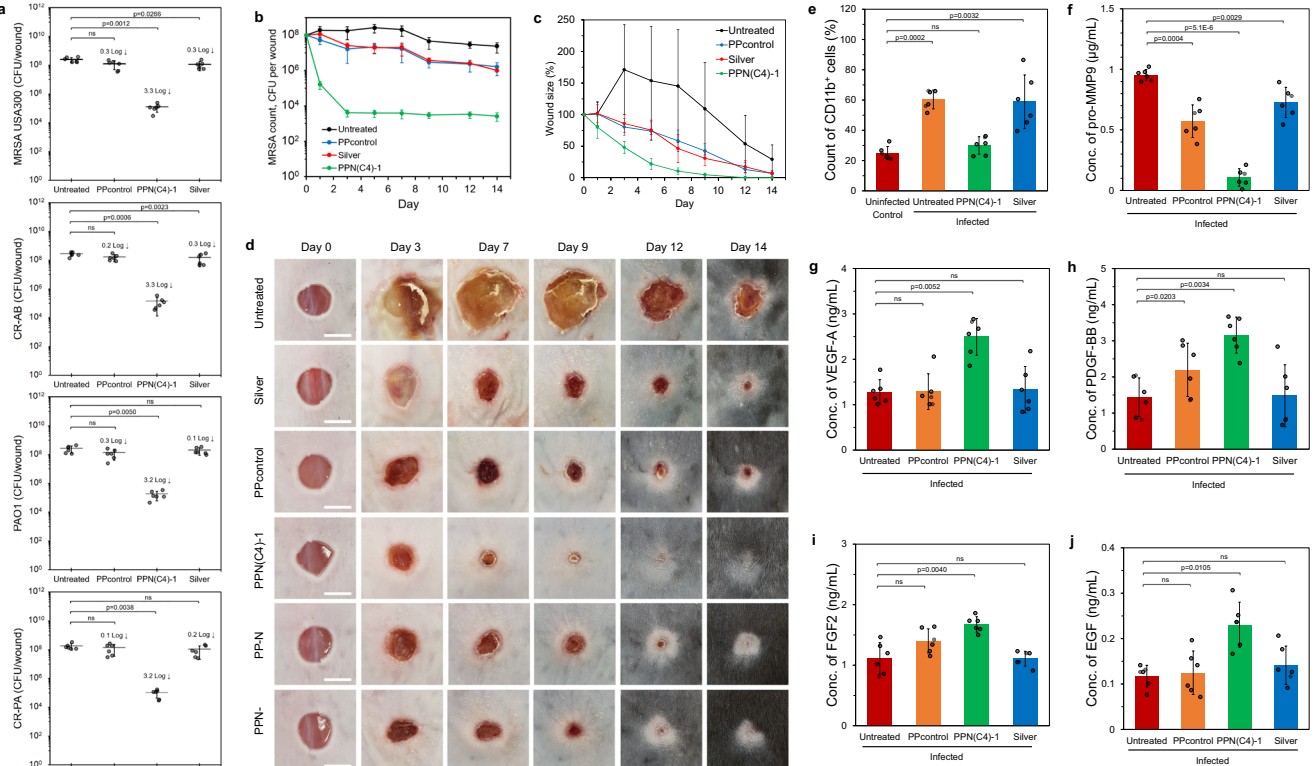

**Fig. 5 | Mouse in vivo diabetic wound infection model with hydrogel films treatment beginning 24 h post-infection. a** Bacterial counts of MRSA USA300, PA01, CR-AB, and CR-PA on various control and treated wounds after 24 h of treatment (n = 6 mice, two-tailed Student's t test). **b–d** Full wound healing study. **b** Bacterial counts of MRSA USA300 on untreated control, silver dressing-, PPcontrol- and PPN(C4)−1 hydrogel-treated wounds on days 0, 1, 3, 5, 7, 9, 12 and 14 post-treatment (n = 6 mice). **c** Wound sizes of untreated control, silver dressing-, PPcontrol- and PPN(C4)−1 hydrogel-treated wounds on various days as a percentage of the initial wound size (n = 6 mice). **d** Visual appearance of representative

untreated control-, silver dressing-, PPcontrol- and PPN(C4)−1 hydrogel-treated wounds between dressing changes. Scale bar = 5 mm. **e–j** Characterization of wound tissues. Measurements in MRSA USA300-infected diabetic mice (n = 6 mice, two-tailed Student's t test) on day 2 post-treatment: **e** Percentage of CD11b⁺ cells in wounds. The percentage of CD11b⁺ cells is directly proportional to the extent of inflammation in the skin. **f** Concentration of pro-MMP9 in wounds. Concentrations of wound healing factors (**g**) VEGF-A, (**h**) PDGF-BB, (**i**) FGF-2 and (**j**) EGF in wounds. Data are presented as mean values ± SD.

bacteria in the wounds was observed during the first 3 days of treatment with PPN(C4)-1 hydrogel, after which the bacterial counts remained consistently low (Fig. 5b). In contrast, the silver dressing and PPcontrol resulted in significantly fewer bacteria being eliminated (less than 1 log reduction), and the untreated control showed almost no reduction in bacteria over 2 weeks (Fig. 5b). Furthermore, the wounds treated with the PPN(C4)-1 hydrogel were smaller, with minimal evidence of slough, than the untreated control, PPcontrol and silver dressing-treated lesions at all time points (Fig. 5c, d, biological replicates are shown in Supplementary Fig. 10). Significantly, the wounds treated with PPN(C4)-1 fully closed at day 12. In contrast, the untreated, PPcontrol and silver-treated wounds failed to close even after 2 weeks. The presence of pus and slough was noticeable on the untreated control wounds, signaling biofilm formation and persistent inflammation throughout the study (Fig. 5d, Supplementary Fig. 10a). The untreated control wounds showed signs of deterioration and displayed evidence of re-infection (Supplementary Fig. 10a).

We further used fluorescence-activated cell sorting (FACS) to determine the percentage of CD11b⁺ cells (i.e., leukocytes, which include monocytes, neutrophils, granulocytes and macrophages) in wounds after 2 days of treatment. Infected untreated (control) wounds contained elevated populations of granulocytes (Fig. 5e, Supplementary Fig. 11). However, infected wounds treated with the PPN(C4)-1 hydrogel did not exhibit an increase in inflammatory (CD11b⁺) cells beyond the levels observed in the control uninfected wounds, indicating that the PPN(C4)-1 hydrogel wound dressing likely attenuates the influx of inflammatory cells resulting from infection by eliminating

bacteria from the wound site. In contrast, treatment with silver dressing did not modulate the number of CD11b⁺ cells in wounds (Fig. 5e, Supplementary Fig. 11), likely attributable to its inefficacy in bacterial removal (Fig. 5a).

Enzyme-linked immunosorbent assay (ELISA) was used to determine the concentration of relevant wound healing factors present in wound exudates after 2 days of treatment. The concentration of pro-MMP9 (precursor of MMP9), which is detrimental to wound healing[36], was high in exudates from the infected untreated control, PPcontrol and silver dressing-treated wounds (Fig. 5f). Interestingly, exudates from wounds treated with the PPN(C4)-1 hydrogel contained a significantly reduced level of pro-MMP9 (Fig. 5f). Furthermore, the concentrations of various wound healing factors (VEGF-A, PDGF-BB, FGF-2 and EGF) measured by ELISA were found to be significantly higher in PPN(C4)-1-treated wounds than in the untreated infected control, PPcontrol and silver dressing-treated wounds (Fig. 5g–j).

We also studied the in vivo infected wound healing effects of PPN- and PP-N hydrogels. Compared with PPN(C4)-1, PPN- (which does not contain NAC in the network) exhibited slower closure of MRSA-infected wounds from day 7 onwards (Fig. 5d), although both hydrogels have the same amount of PIM(C4)-Mal incorporated. This might be attributed to the lack of NAC in PPN-, which corroborates the decreased expression of growth factors (specifically VEGF-A, PDGF-BB and FGF-2) in wound exudates compared to PPN(C4)-1 (Supplementary Fig. 12). Additionally, without the presence of PIM(C4)-Mal in the network, PP-N does not perform as well as PPN(C4)-1 in helping the closure of MRSA-infected wounds from day 7 onwards (Fig. 5d) due to the

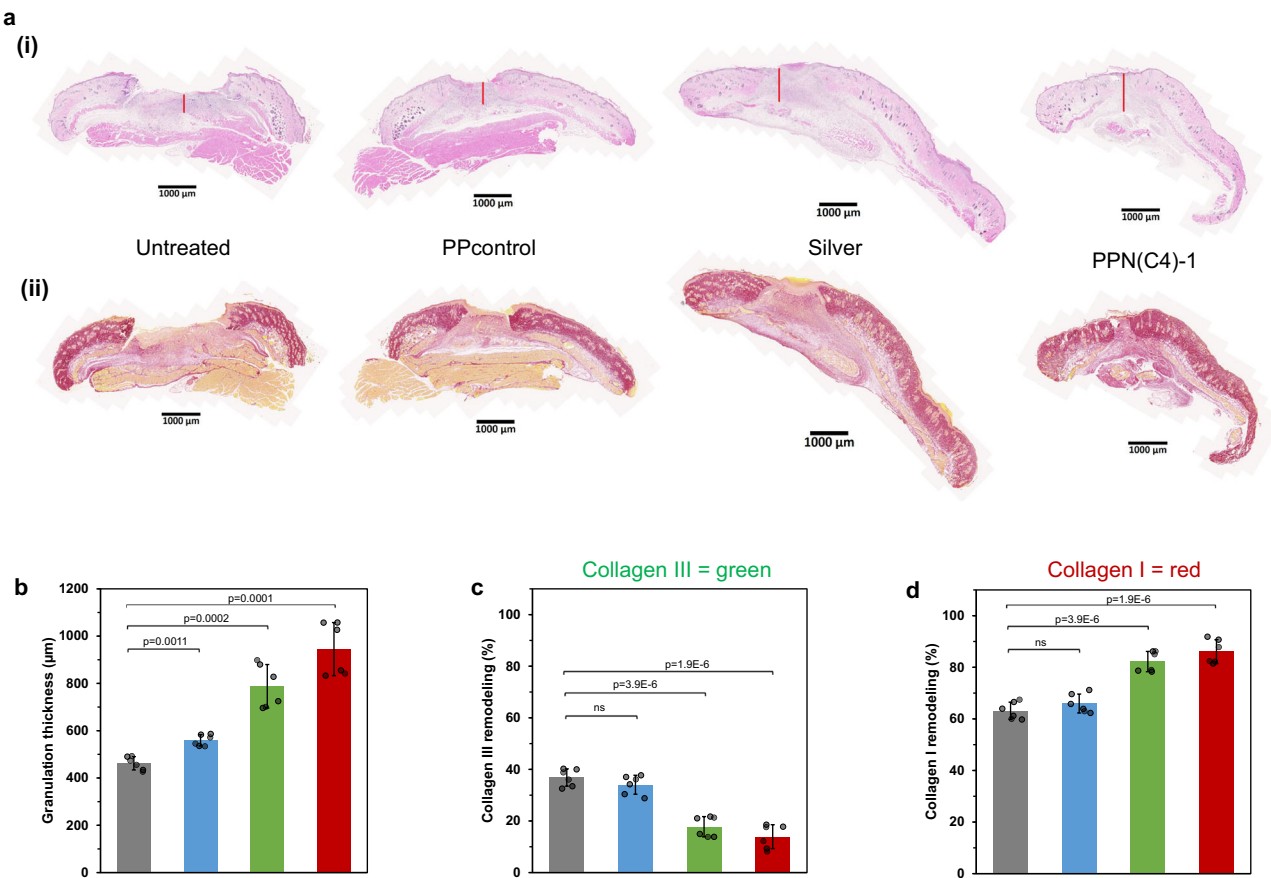

**Fig. 6 | Wound histology of diabetic mouse model treated 24 h post-infection of MRSA USA300 with hydrogel films. a** Representative images depicting (i) H&E and (ii) picrosirius red stained tissue sections of untreated control, silver dressing, PPcontrol, and PPN(C4)−1 hydrogel-treated wounds on day 7 post-treatment. Scale bar = 1 mm. **b** Quantification of granulation tissue formation thickness based on histological wound samples harvested on day 7 post-treatment. Quantification of (**c**) collagen III (immature collagen, stained green) and (**d**) collagen I (native collagen, stained red) deposition and remodeling in the histological wound samples harvested on day 7 post-treatment. (*n* = 6 mice, two-tailed Student's *t* test, data are presented as mean values ± SD).

lack of antimicrobial functionality. Wounds treated with PP-N also exhibited reduced expression of EGF growth factor, although not VEGF-A, PDGF-BB and FGF-2, which may be attributed to a too early time point of measurement (2 days of treatment, Supplementary Fig. 12). These results demonstrate that the dual-function hydrogel PPN(C4)-1 facilitates faster wound closure and increased expression of growth factors compared to hydrogels with only one or the other of the bioactive component.

After 7 days of treatments, we conducted H&E histological assessments on the wound tissues. Compared to the untreated and PPcontrol-treated wounds, both the silver and PPN(C4)-1 hydrogel-treated wounds demonstrated significantly increased thickness of granulation tissues (marked with a vertical red line in Fig. 6a(i)), indicating enhanced healing with improved keratinocyte and fibroblast proliferation (Fig. 6b). Thicker granulation tissue, which is composed of proliferating fibroblasts, blood vessels, and extracellular matrix, is often observed during the normal wound healing process[37]. The granulation tissue provides a scaffold for keratinocyte migration and proliferation[38]. The presence of a thicker granulation tissue suggests an active wound healing process, which may indicate improved keratinocyte proliferation. Moreover, the PPN(C4)-1 hydrogel-treated wounds displayed more developed epithelial tongues compared to the other sample groups (Fig. 6a(i), Supplementary Fig. 13), indicating heightened proliferation and migration of keratinocytes. This observation further emphasizes the positive impact of PPN(C4)-1 hydrogel treatment on wound healing. We also

analysed the collagen remodeling in wound tissues after 7 days of treatments by picrosirius red staining (Fig. 6a(ii), Supplementary Fig. 14). In the untreated and PPcontrol hydrogel-treated wounds, collagen type III, which is immature and forms during the initial healing process to close and protect the wound from external environments, comprised approximately 33-37% of the total collagen (Fig. 6c). Conversely, in the silver and PPN(C4)-1 hydrogel-treated wounds, most of the collagen type III had been replaced by the more matured collagen type I, comprising 82-86% of the overall collagen (Fig. 6d). This shift towards more mature collagen in the PPN(C4)-1 hydrogel-treated wounds indicates not only enhanced wound closure but also the essential maturation of the wound tissues, promoting overall regeneration of healthy skin.

In summary, the ex vivo 3D skin wound model demonstrated the good biocompatibility of PIM(C4)-Mal-based hydrogels in reconstructed human skin tissue and the promotion of wound healing (even without infection) by NAC. The addition of covalently linked PIM(C4) to the hydrogels did not cause significant toxicity and did not affect the biocompatibility of the PEG-derived hydrogels. In the infected wounds of diabetic mice, the dual-component PPN(C4)-1 hydrogel significantly reduced (>3 log reduction) Gram-positive and Gram-negative MDR biofilm bacteria. Furthermore, PPN(C4)-1 accelerated the closure of MRSA-infected wounds in the murine diabetic model and caused higher expression of wound healing factors, such as VEGF-A, PDGF-BB, FGF-2 and EGF, than with the silver dressing, PPcontrol or the untreated infected control conditions (Fig. 5b–j). The eradication

**Table 2 | Compositions of the Alg-PPN(C4) fibers in 1 mL of DI water**

| Fiber formulation | Alg-SH | PIM(C4)-Mal | PEG-2Mal-2NAC | |
| --- | --- | --- | --- | --- |
| | | | PEG-4Mal | NAC |
| Alg | 50 mg | – | – | – |
| Alg-PPN(C4)–0.1 | 50 mg | 0.1 mg | 10 mg (0.5 µmol) | 0.16 mg (1 µmol) |
| Alg-PPN(C4)–1 | 50 mg | 1 mg | 10 mg (0.5 µmol) | 0.16 mg (1 µmol) |
| Alg-PPN(C4)-5 | 50 mg | 5 mg | 10 mg (0.5 µmol) | 0.16 mg (1 µmol) |
| Alg-PPN(C4)–10 | 50 mg | 10 mg | 10 mg (0.5 µmol) | 0.16 mg (1 µmol) |

of bacteria in the in vivo diabetic mouse study occurred in the earlier time periods (first 3 days) of treatment with PPN(C4)-1. Thus, the addition of both PIM(C4) and NAC is important for the removal of bacteria and wound healing, respectively.

As many chronic wounds are rather deep and film format dressing is not ideal for such wounds, we also produced alginate hydrogel fibers which are covalently linked with PPN, that can fill deep wounds and conform to their surfaces. A series of fiber formulations were prepared using Alg-SH, PEG-4Mal, PIM(Cn)-Mal ($n$ = 4, 8, or 10), and NAC in DI water (Table 2). These hydrogel fibers are labeled as Alg-PIM(Cn)-PEG-NAC-x (Alg-PPN(Cn)-x), where the suffix (x) refers to the PIM(Cn)-Mal concentration (x mg/mL) (Table 2). Linear PIM(Cn)-Mal with different alkyl (Cn) linkers ranging from butane (C4) to decane (C10) were pre-synthesized and the PIM(Cn) solutions were screened (Supplementary Fig. 1). The different PIM(Cn) solutions have different antibiofilm effects (Supplementary Fig. 15); in addition to the solution forms of PIM(C4), PIM(C8) and PIM(C10), they were further made into Alg-PPN fiber for testing.

Briefly, the 4-arm PEG-4Mal was pre-modified with 2 equivalents of NAC to convert, on average, 2 PEG arms per chain to NAC terminals to produce PEG-2Mal-2NAC (Fig. 1c, Supplementary Fig. 2a). Thiol-modified alginate (Alg-SH) was pre-reacted with PIM(Cn)-Mal to produce Alg-SH-PIM(Cn) (Fig. 1c, Supplementary Fig. 2b). Some thiol groups remained unreacted in Alg-SH-PIM(Cn) for subsequent click reactions, and were quantified (Supplementary Fig. 16). The Alg-SH-PIM and PEG-2Mal-2NAC were further reacted via the thiol-maleimide reaction to form the Alg-PPN solution (Fig. 1c), which was then coagulated in a CaCl₂ bath to obtain Alg-PPN fibers. An alginate fiber control (labeled as Alg) was produced by extruding Alg-SH solution into a CaCl₂ bath (Table 2). The as-prepared fibers were washed thoroughly in DI water with sonication and dehydrated in ethanol. The hydrogel fibers showed a range of tensile strengths of 45 - 62 kPa and tensile strains (elongation) of 27 - 34% (Fig. 2c(ii), d(ii)). The amount of leachable PIM, NAC, and PEG components after washing was found to be less than 2.1%, 1.2%, and 1.2% of the initial PIM, NAC, and PEG components respectively (Supplementary Figs. 17-18, Supplementary Table 3–5). We also used a commercial alginate fiber (labeled as Alg-Com) and commercial silver-containing alginate fiber (Alg-Ag) as control samples.

Contact killing efficacies of the Alg-PPN and control fibers were measured in vitro against various MDR Gram-negative and Gram-positive bacteria, specifically MRSA USA300, PAO1, CR-PA and CR-AB. The Alg-Com and control Alg fibers did not exhibit substantial bactericidal properties (Fig. 7a–d). The Alg-PPN(Cn)-5 and Alg-PPN(Cn)−10 fibers totally eradicated (in 1 h) Gram-positive MRSA USA300 and CR-AB, as well as Gram-negative PAO1 and CR-PA bacteria, which were inoculated onto the fibers (Fig. 7a–d, Supplementary Tables 6–8). The Alg-PPN(Cn)-0.1 and Alg-PPN(Cn)-1 fibers did not completely eradicate the bacteria (Fig. 7a–d), probably due to the lower concentration of the active antibacterial PIM(Cn)-Mal tethered to the fibers.

The antibiofilm efficacies of fibers towards preformed biofilms were also studied. The Alg-Com and control Alg fibers did not exhibit antibiofilm activity towards the tested Gram-positive and Gram-negative bacteria (Fig. 7e–h). Alg-PPN(C4) exhibited a range of 1.39–3.81 log reductions towards MRSA biofilm (Fig. 7e,

Supplementary Table 9) due to the presence of the PIM(C4)-Mal active component in the fibers. Alg-PPN(C8), which has more alkyl carbon atoms in its PIM component, exhibited a higher log reduction range against MRSA biofilms of 1.84–6.80, depending on the PIM(C8) concentration (Fig. 7e, Supplementary Table 10). Alg-PPN(C10)−5 and Alg-PPN(C10)-10 fibers eradicated MRSA biofilms completely (Fig. 7e, Supplementary Table 11). This trend was also observed in the antibiofilm assays towards CR-AB, PAO1, and CR-PA bacteria (Fig. 7f–h, Supplementary Tables 9–11), demonstrating the potency of Alg-PPN in combating various biofilm bacteria.

The in vitro biocompatibility of the fibers against 3T3 mouse fibroblasts and human dermal fibroblasts (HDFs) was studied with supernatant extracts and direct contact with Alg-PPN(Cn) fibers. The viabilities of HDF and 3T3 cells exposed to the extracts of Alg-PPN(C4) fibers were high, in the range of 92 - 99% and 95 - 98%, respectively (Supplementary Fig. 19a, b). These results suggest that the Alg-PPN(C4) fibers are cationic low-leaching. After contact incubation with the Alg-PPN(C4) fibers, the cell viabilities of HDF and 3T3 fibroblasts were in the range of 85 - 91% and 88 - 93%, respectively (Supplementary Fig. 19a, b), indicating low acute cytotoxicity of the fibers. The viabilities of 3T3 cells in contact with Alg-PPN(C8)-x (for x ≤ 5) and Alg-PPN(C10)-x (for x ≤ 1) were >70% (Supplementary Fig. 19c, d).

In the above in vitro analyses, Alg-PPN(C8)-5 achieved at least 4 log reduction of the MDR Gram-positive and Gram-negative biofilm bacteria with low cytotoxicity. We therefore chose this formulation for in vivo validation studies using mice. We investigated the healing of Gram-negative CR-PA-infected wounds on diabetic mice treated with control Alg, silver-containing commercial alginate (Alg-Ag), and Alg-PPN fibers over a 2-week period. For the Alg-PPN treatment group, Alg-PPN(C8)-5 (containing PIM(C8)) was applied to the wound until day 5 to eradicate bacteria, followed by replacement with the more biocompatible Alg-PPN(C4)-5 (containing PIM(C4)) from day 5 to day 14. The largest reduction in the CR-PA bacterial count was observed in wounds treated with Alg-PPN(C8)-5 fiber during the initial 5 days (Fig. 8a). Wounds treated with the Alg-PPN fibers were smaller, with minimal evidence of slough, than the untreated control, Alg, and Alg-Ag fibers-treated wounds at all time points (Fig. 8b, c, Supplementary Fig. 20). Pus and slough were observed on the untreated control wounds, indicative of biofilm formation and sustained inflammation (Fig. 8c, Supplementary Fig. 20a). As in the corresponding PPN hydrogel tests, the untreated control wounds deteriorated and exhibited evidence of re-infection. After a 2-week period, the wounds fully closed for the Alg-PPN-treated group but did not close for the other groups (Fig. 8c, Supplementary Fig. 20).

To evaluate the effect of dressing treatments on immune cell infiltration, we characterized populations of immune cells in wound tissues after 2 days of treatment. The untreated infected wounds exhibited the highest percentage of CD11b⁺ inflammatory cells (Fig. 8d). Infected wounds treated with Alg-PPN fiber exhibited the lowest percentage of CD11b⁺ cells, followed by wounds treated with Alg-Ag fiber (Fig. 8d, Supplementary Fig. 21b–e). The low percentage of CD11b⁺ cells indicates that fewer leukocytes infiltrated wounds to fight infection in wounds treated with Alg-PPN fiber, which thus caused lower inflammatory response in the wound tissues. Most of these

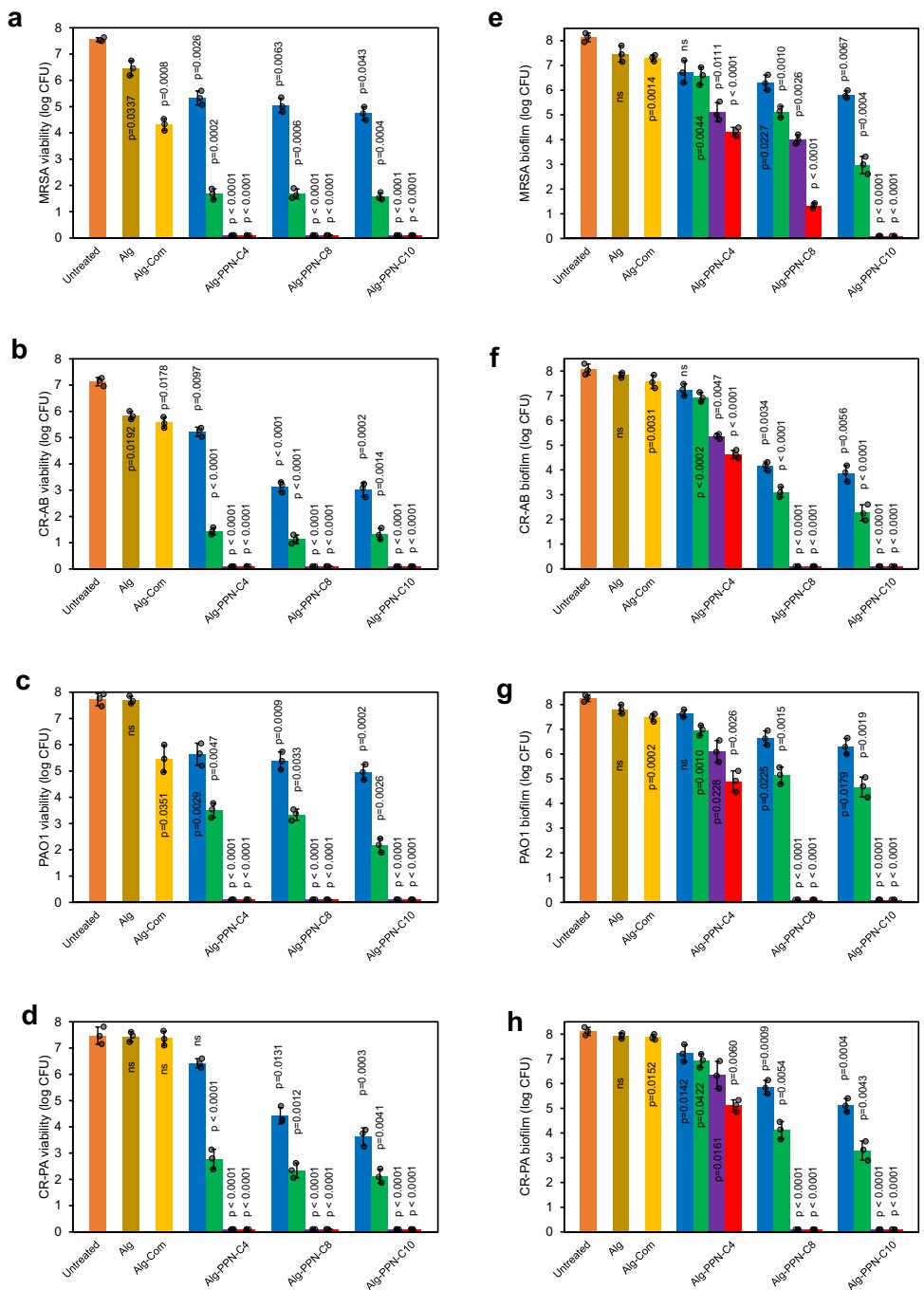

**Fig. 7 | Antibacterial and antibiofilm activities of the fiber hydrogels.** Viability of (**a**–**d**) planktonic bacteria and (**e**–**h**) biofilms of (**a**, **e**) MRSA, (**b**, **f**) CR-AB, (**c**, **g**) PAO1, and (**d**, **h**) CR-PA after contact incubation with the surface of Alg-PPN(Cn)-0.1 (blue), Alg-PPN(Cn)−1 (green), Alg-PPN(Cn)-5 (purple), and Alg-PPN(Cn)−10 (red) fibers at 37 °C ($n = 3$ biologically independent samples, two-tailed Student's $t$ test, $p$ values denote significant difference compared to untreated controls, data are presented as mean values ± SD). The contact periods were 1 h for planktonic bacteria and 24 h for biofilms.

CD11b[+] cells in the Alg-PPN fiber treatment group were Ly6G neutrophils (Fig. 8d, Supplementary Fig. 21a), suggesting reduced acute inflammatory responses in the wound tissues. (Ly6G is a specific marker for mouse neutrophils.) The population of Ly6G cells on the Alg-PPN-treated wounds was also lower than that on the Alg-Ag-treated wounds (Supplementary Fig. 21a). These results suggest the efficacy of Alg-PPN over Alg-Ag in the killing of wound site bacteria, which alleviates the bioburdens on infected wounds for accelerated healing. Neutrophils are among the earliest immune cells to arrive at wound sites. Thus, for the untreated and Alg control groups, high percentages

of CD11b[+] cells but low Ly6G-positive neutrophils suggest that the inflammation has advanced beyond the early phase.

Another consequence of the acute infiltration of immune cells, especially neutrophils, is the release and activation of proteases. We assayed the concentration of pro-MMP9 (a specific macrophage product[39] and precursor to MMP9[36]) using ELISA and detected the highest concentrations in control untreated wounds due to over-production of the innate response system to fight the high amount of bacteria. Wounds treated with the Alg-PPN fiber exhibited a significantly reduced level of pro-MMP9 (Fig. 8e), indicating the high

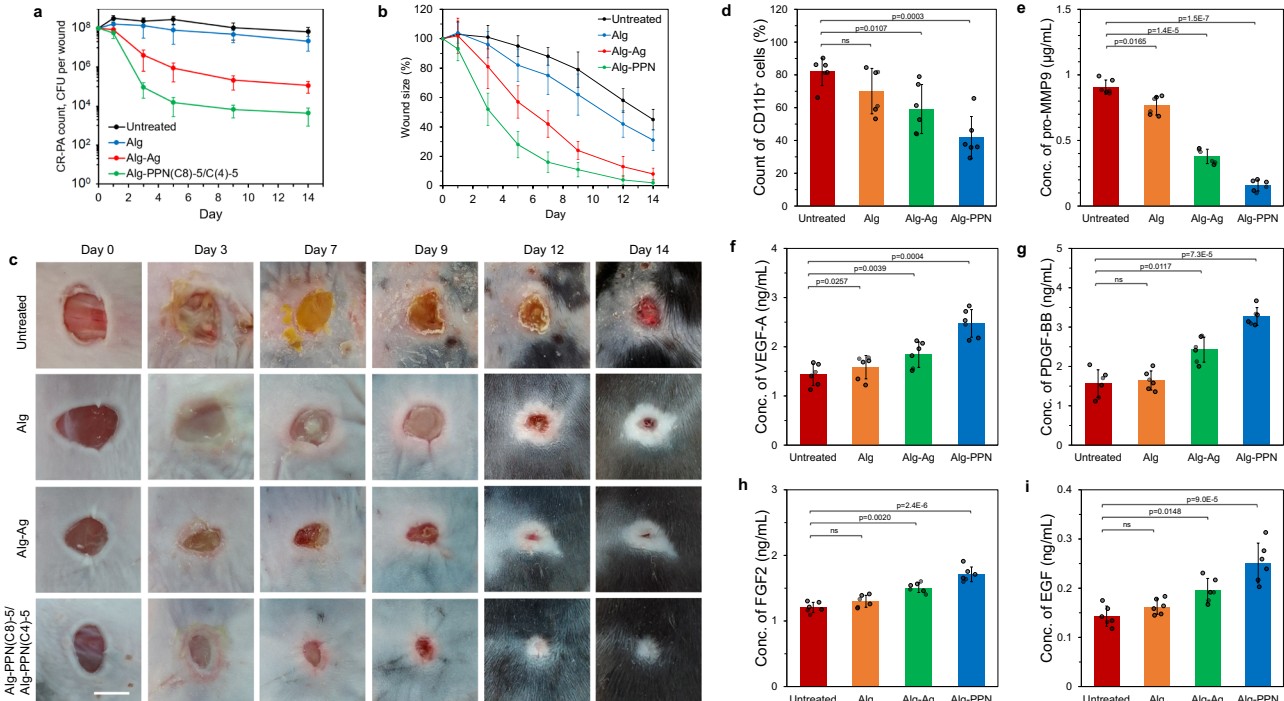

**Fig. 8 | Full wound healing study of murine infected diabetic wounds with hydrogel fibers treatment beginning 24 h post-infection. a** Bacterial counts of CR-PA on untreated control, Alg, Alg-Ag, and Alg-PPN fibers-treated wounds on days 0, 1, 3, 5, 7, 9, 12 and 14 post-treatment (n = 6 mice). **b** Wound sizes of untreated control, Alg, Alg-Ag, and Alg-PPN fibers-treated wounds on various days as a percentage of the initial wound size (n = 6 mice). **c** Visual appearance of representative untreated control, Alg, Alg-Ag, and Alg-PPN fibers-treated wounds between dressing changes. Scale bar = 5 mm. **d–i** Characterization of wound tissues. Measurements in CR-PA-infected diabetic mice (n = 6 mice, two-tailed Student's t test) on day 2 post-treatment: **d** Percentage of CD11b+ cells in wounds. The percentage of CD11b+ cells is directly proportional to the extent of inflammation in the skin. **e** Concentration of pro-MMP9 in wounds. Concentrations of wound healing factors (**f**) VEGF-A, (**g**) PDGF-BB, (**h**) FGF-2 and (**i**) EGF in wounds. Data are presented as mean values ± SD.

efficacy of Alg-PPN in killing bacteria. Wounds treated with the Alg-Ag fiber also exhibited significantly reduced pro-MMP9 concentrations, although to a lesser degree than Alg-PPN. The concentrations of several major wound healing factors (VEGF-A, PDGF-BB, FGF-2 and EGF) were assayed to be the highest in Alg-PPN fiber-treated wounds (Fig. 8f–i), correlating with faster healing rate in this group than in untreated control, Alg, and Alg-Ag-treated wounds.

## Discussion

Diabetic wounds are commonly chronically inflamed due to bacterial infection which often leads to elevated reactive oxygen species (ROS) levels[40]. These factors conspire to impair the genesis and repair of blood vessels, reduce the biosynthesis and delivery of wound healing factors, and cause failure in the management of the bioburden on wounds. Herein, we have developed a hydrogel (called PPN) that contains 2 bioactive components which provide synergistic functions to alleviate infection and quench the excessive ROS to result in accelerated closure of infected diabetic wounds. Further, our *N*-acetylcysteine (NAC) component (even in uninfected wound model) directly promote keratinocyte proliferation and the squamous differentiation of keratinocytes, to result in thicker re-epithelialization.

In our previous work on cationic hydrogels that are antibacterial but not wound healing[41], we discovered that cationic hydrogels kill bacteria by (i) absorbing them into their pore spaces (via the hydrodynamic drag force generated by the evaporation of water from the hydrogel and its subsequent rehydration), followed by (ii) contact killing of bacteria by the cationic polymers of the pore walls. The cationic hydrogel here similarly absorbs bacteria into its pore spaces and then contact kill them via the cationic hydrogel pore walls[42].

The accelerated wound healing mechanism of our dual functionality hydrogel stems from the synergy of bacteria removal by the crosslinked cationic hydrogel which itself results in reduced inflammatory response, together with the antioxidative property of NAC. Firstly, our PPN hydrogel dressing eradicates and eliminates bacteria from the wound site. The bacterial removal from the site results in a significant reduction in the influx of inflammatory cells typically associated with infection, as evidenced by the suppressed level of CD11b+ cells in the infected mice wounds treated with PPN(C4)-1 hydrogel as compared to the untreated wounds (Fig. 5e, Supplementary Fig. 11). Secondly, the NAC reduces the levels of ROS which are high in chronic wounds[43,44], as they diffuse into the PPN hydrogel and are quenched. NAC, a precursor to glutathione (GSH), can substitute for GSH function in tissues. The NAC antioxidant neutralizes free radicals, reduces oxidative stress and inflammation, and boosts the immune system[45], so that it reduces inflammation and tissue damage due to high ROS levels, and allow for progression to normal healing. Thirdly, the PPN hydrogel promotes the supply of wound healing factors to the wound sites, as shown by the significantly higher concentrations of VEGF-A, PDGF-BB, FGF-2 and EGF in the PPN(C4)-1 hydrogel-treated mice wounds than in all other treatment groups (Fig. 5g–j). This effect can be attributed to the suppression of inflammatory mediators, such as TNF-alpha and IL-1, stemming from the reduced presence of inflammatory cells and biofilm bacteria.

Finally, the incorporation of the antioxidant NAC in the PPN hydrogel promotes keratinocyte differentiation, as substantiated using the ex vivo 3D de-epidermised dermis human skin equivalent (DED-HSE) model (that has repopulated allogeneic donor keratinocytes). Uninfected wounds treated with PP-N (hydrogel without PIM

but with NAC) exhibited a 35% increase in epidermal thickness when compared to wounds treated with the PPcontrol (Supplementary Table 2), indicative of the enhanced re-epithelialization effect attributed to NAC incorporation. Additionally, when compared to the PPcontrol group, uninfected wounds treated with PP-N and PPN(C4)-1 hydrogels exhibited significantly stronger signal intensity for K14 and K10 proteins, respectively (Fig. 4c and Supplementary Table 2), indicating that NAC incorporation into the composite PPN hydrogel enhances the squamous differentiation of keratinocytes. Consequently, the combination of NAC and the antibacterial polymer within the PPN hydrogel operates synergistically to promote wound healing by collectively diminishing ROS levels and inflammation.

Our PPN(C4)-1 hydrogel is more bactericidal to biofilm bacteria than the commercial silver-based wound dressing, even to carbapenem-resistant *P. aeruginosa* and *A. baumannii* (CR-PA and CR-AB), which urgently need new antibacterial therapies[32]. Furthermore, wounds treated with the PPN(C4)-1 hydrogel exhibited minimal erythema, suppressed inflammation, and accelerated wound closure. The diabetic murine model results confirmed that the hydrogels with PIM and NAC combination are the best formulation for the removal of biofilms and acceleration of wound healing, as treatment with PPcontrol (without PIM and NAC) did not significantly kill bacteria, and the wounds healed more slowly (Fig. 5d). The hydrogels formulated with only a single active component (i.e., either PP-N or PPN-) exhibited slower healing and reduced levels of wound healing factors (Fig. 5d, Supplementary Fig. 12). Many studies on wound dressings only focus on one aspect of wound healing or one type of wound (*e.g.*, infected wounds or diabetic wounds)[46–49], but our hydrogel has dual functionalities that lead to improved healing of infected wounds, fulfilling the unmet need.

Dual-function antibacterial and antioxidative features are considered to be the most effective current solution for wound treatment. Several such dual-functional formulations have been reported but they suffer from being grossly leaching, or not so effective particularly against resistant bacteria[50–53]. Zhou et al. developed a semi-interpenetrating polymer network (sIPN) hydrogel with antibacterial and anti-inflammatory activities that could degrade completely within 2 h after being applied to wounds with in vivo murine model[50]. Although the sIPN hydrogel was fairly effective in killing non-resistant *E. coli* and *S. aureus* (less than 2 log reduction), its antibacterial effect which is due to degradation of the hyaluronic acid network, releases the foreign polyvinyl with dangling cationic imidazolium moieties onto the wound site. This degradable hydrogel is not a retrievable dressing, and large amounts of degraded polymers are left on the wound. Romero-Montero et al. developed hydrogel nanofibers loaded with the antibiotic clindamycin and the antioxidant poly(gallic acid)[51]. However, the release of antibiotics from this wound dressing product raises concern for the spread of bacterial resistance through horizontal gene transfer. Other groups have developed xerogel film[52] and hydrogel membrane[53] that can rapidly (within 1 or 2 h) release quercetin, a plant flavonoid that was reported to have antibacterial and antioxidant properties[54]. However, these gels have been demonstrated to kill only non-resistant bacteria in vitro or treat uninfected wounds of non-diabetic mice. Considering these recent studies, our standalone and low-leaching PPN hydrogel film and fiber achieved killing broad-spectrum multidrug-resistant (MDR) bacterial biofilms, as well as accelerating the closure of biofilm-infected diabetic wounds, which have not been demonstrated previously.

We employed the facile crosslinking chemistry of thiol-maleimide Michael addition reaction. We found no evidence that bacterial extracts (Supplementary Fig. 6a, b) or wound fluids (Supplementary Fig. 6c) degrade PPN(C4)-1 hydrogel, nor did we find evidence for degradation of PPN(C4)-1 hydrogel film and Alg-PPN(C8)-5 hydrogel fiber on wounds (Supplementary Fig. 7, Supplementary Movies 1–4). The hydrogel's overall structural integrity is well preserved.

The hydrogels can be assembled into multiple formats; we have created film and fiber formats which are suitable for use in treatment of shallow and deep wounds, respectively. The composite hydrogel film demonstrates effective biofilm removal and accelerates healing of infected cutaneous wounds in diabetic mice. With the alginate fibers (which are suitable for deep, as well as shallow, wounds), the bioactive PIM and NAC components retained their antibacterial and antioxidative properties in murine wound models. This technology may alleviate the growing problem of chronic and diabetic wounds, as current treatments are limited by their contraindications. Furthermore, hydrogels can also be assembled by electrospinning or 3D printing to easily create custom formats or shapes suited to fit specific applications. Finally, this hydrogel can also be used in other biomedical applications, such as coatings for biomedical devices.

In summary, our study introduces a crosslinked hydrogel with two covalently bound, ultralow-leachable bioactive components: the highly potent antibacterial cationic PIM and the antioxidative NAC. The composite hydrogel exhibits efficient and broad-spectrum biofilm removal capabilities, while also accelerate the healing process in infected diabetic wounds. It is intrinsically antibacterial and antioxidative and does not require other processes such as photothermal irradiation so that this standalone dressing is easy and safe to use. Unlike previous dual functionality dressings, the crosslinked hydrogel is completely devoid of leachable antibiotics, metal compounds, carbon nanotubes or nanoparticles. This ultra-low leaching feature distinguishes it from many other drug-releasing wound dressings, promoting a safer and more biocompatible alternative. It has good structural integrity and is easy to be removed cleanly from the wounds. Furthermore, different formats such as films and fibers can be made to conform to wounds. These advantages greatly enhance patient comfort and support an uncomplicated healing process.

## Methods
### Materials
1,4-diaminobutane, 1,6-diaminohexane, 1,8-diaminooctane, 1,10-diaminodecane, formaldehyde (37%), glyoxal (40%), acetic acid, maleic anhydride, sodium bicarbonate, *N*-acetylcysteine (NAC), alginic acid sodium salt (sodium alginate), L-cysteine (97%), *N*-ethyl-*N'*-(3-dimethylaminopropyl)carbodiimide (EDC, 97%), sodium hydroxide (NaOH), sodium chloride (NaCl), calcium chloride (CaCl$_2$), hydrochloric acid (HCl), lecithin, Tween 80, sodium thiosulfate, 5,5-dithio-bis(2-nitrobenzoic acid) (DTNB, 99%), and 3-(4,5-dimethylthiazol-2-yl)-2,5-diphenyltetrazolium bromide (MTT, 98%) were purchased from Sigma-Aldrich Corp. (St. Louis, MO). Poly(ethylene glycol) tetra thiol (PEG-4SH, $M_n$ 20,000 Da) and poly(ethylene glycol) tetra maleimide (PEG-4mal, $M_n$ 20,000 Da) were purchased from Biochempeg Sci. Inc. (Watertown, MA). Fluorescein-5-maleimide (F5M), phosphate-buffered saline (PBS), Mueller-Hinton broth (MHB), tryptic soy broth (TSB), Luria-Bertani (LB) agar, ELISA kits for mouse pro-MMP9 (EMMMP9), VEGF-A (BMS619-2), PDGF-BB (BMS2071), FGF-2 (EMFGF2), and EGF (EMEGF) were purchased from Thermo Fisher Sci. Inc. (Waltham, MA). CD11b$^+$ (130-113-231) and Ly6G (130-102-296) antibodies were purchased from Miltenyi Biotec. Mouse monoclonal anti-K10 (DE-K10) was purchased from Dako (DKO.M7002). Mouse monoclonal anti-K14 (LL001) was a supernatant from E. Birgitte Lane's lab (A*STAR). Mouse monoclonal anti-p63 (4A4) was purchased from Abcam (Ab735). *Enterococcus faecium* (19434), *Enterobacter cloacae* (13047), *Klebsiella pneumonia* (13883), methicillin-resistant *Staphylococcus aureus* (MRSA USA300, BAA-40, and LAC), carbapenem-resistant *Acinetobacter baumannii* (CR-AB), *Pseudomonas aeruginosa* 01 (PAO1), carbapenem-resistant *P. aeruginosa* (CR-PA), and standard 3T3 mouse fibroblast cells were purchased from American Type Culture Collection (ATCC, Manassas, VA). Human dermal fibroblast cells (HDFs, NHDF-Ad, CC-2511) were purchased from Lonza (Basel,

Switzerland). Ethanol and dimethyl sulfoxide (DMSO) were of analytical grade.

## Synthesis of polyimidazolium containing maleimide terminal groups (PIM(Cn)-Mal)

PIM(C4)-Mal was synthesized via the following procedure. 2.974 g (33.7 mmol) of 1,4-diaminobutane was dissolved in 75 mL of acetic acid and cooled in an ice bath. 2.735 g (33.7 mmol) of 37% formaldehyde solution and 4.895 g (33.7 mmol) of 40% glyoxal solution were dissolved in 37.5 mL of DI water, and the mixture was added dropwise to the 1,4-diaminobutane solution. The resulting mixture was stirred at 25 °C for 24 h, after which the solvents were evaporated with a rotary evaporator. The dried product was re-dissolved in DI water, dialyzed in DI water for 3 days using regenerated cellulose membrane (MWCO 1 kDa), and lyophilized to obtain PIM(C4). 2 g (0.7 mmol) of PIM(C4) was dissolved in 50 mL of acetic acid, to which 0.27 g (2.8 mmol) of maleic anhydride and 0.24 g (2.8 mmol) of sodium bicarbonate were added. This solution was stirred at 100 °C for 24 h, after which the solvent was removed with a rotary evaporator. The product was re-dissolved in DI water, dialyzed in DI water for 3 days using regenerated cellulose membrane (MWCO 2 kDa), and lyophilized to obtain PIM(C4)-Mal. $^1$H NMR (300 MHz, DMSO) $\delta$: 10.31 (C-H 'a'), 7.94 (C-H 'b'), 6.29 (C-H 'f'), 4.29 (-CH$_2$-, 'c') 1.79 (-CH$_2$-, 'd'), 1.59 (CH$_3$COO-, 'e') (Supplementary Fig. 3a).

PIM(Cn)-Mal with $n$ = 6, 8, and 10 were synthesized following the same method as above, with substitution of the respective diaminoalkanes in place of 1,4-diaminobutane, i.e., using 1,6-diaminohexane for PIM(C6)-Mal, 1,8-diaminooctane for PIM(C8)-Mal, and 1,10-diaminodecane for PIM(C10)-Mal.

## Preparation of the hydrogel films (PPN(Cn))

Hydrogels films were prepared by casting of solutions of the precursor components mixed using a vortex mixer. The antioxidant component solution contained 10% (w/v) of PEG-4SH and 2 mM of NAC in DI water. The antibacterial component solution contained 10% (w/v) of PEG-4Mal and 0.2, 2 or 20 mg/mL of PIM(Cn)-Mal in DI water. These two solutions were mixed at equal volume, and 50 μL aliquots of the mixed solution were quickly deposited in the wells of a 96-well plate. The final composition of the hydrogel was 5% (w/v) PEG-4SH, 5% (w/v) PEG-4Mal, 1 mM NAC and 0.1, 1 or 10 mg/mL PIM(Cn)-Mal (Table 1). The hydrogel samples were kept at 25 °C for 5 min to form PPN(Cn) hydrogels. DI water was then added to the wells to swell the hydrogels. After swelling for 15 min, the hydrogels were washed thrice in ethanol and thrice in DI water in an ultrasonic bath (150 W, 37 kHz) to remove all unreacted precursors. Two PPN(Cn)-derived hydrogels lacking either PIM or NAC were also made: PP-N was made without PIM(Cn), while PPN- was made without NAC (Table 1). A control gel (PPcontrol) with no antioxidant or antibacterial functionalities, i.e. only 5% (w/v) PEG-4SH and 5% (w/v) PEG-4Mal, was also prepared.

## Synthesis of thiol-functionalized alginate (Alg-SH)

Alg-SH was synthesized by using amidation of carboxylic acid group[55,56]. Sodium alginate (2 g, 11.4 mmol of alginate repeat units) was dissolved in 200 mL of DI water and activated by adding EDC (50 mM). The mixture was stirred at 25 °C for 1 h. Next, L-cysteine (1.4 g, 11.4 mmol) was dissolved in 100 mL of DI water and added to the reaction mixture dropwise under stirring. During the addition, the pH of the reaction mixture was monitored and adjusted to 5 using 2 M NaOH solution. The mixture was then stirred at 25 °C for 24 h. The product was dialyzed (MWCO 8 kDa) in DI water containing 1 mM HCl at 25 °C. Next, two consecutive cycles of dialysis were performed in DI water containing 1% NaCl and 1 mM HCl. The sample was lyophilized to obtain the Alg-SH polymer. The product was stored at 4 °C until further use.

## Preparation of PEG-2Mal-2NAC

PEG-4Mal (10 mg, 0.5 μmol) was dissolved in 0.2 mL DI water. Then, 0.1 mL aqueous solution of NAC (0.16 mg, 1 μmol) was added dropwise with vortexing. The mixture was agitated to react for 1 h to form the PEG-2Mal-2NAC polymer. With a PEG-4Mal:NAC design molar ratio of 1:2, the thiol-maleimide click reaction leads to the attachment of 2 molecules of NAC on each PEG-4Mal molecule (statistical average).

## Formation of fibers of alginate modified with PIM-Mal and PEG-2Mal-2NAC (Alg-PPN)

Alg-SH (50 mg, 35.5 μmol of thiol moieties) was dissolved in 0.5 mL DI water. Then, an aqueous solution of PIM(Cn)-Mal (0.1, 1, 5, or 10 mg) in 0.2 mL of DI water was added dropwise with vortexing (Table 2). The mixture was agitated for 1 h to allow reaction between the maleimide group of PIM(Cn)-Mal and the thiol group of Alg-SH, resulting in the Alg-SH-PIM(Cn) polymer.

To further graft PEG-2Mal-2NAC onto Alg-SH-PIM(Cn), 0.3 mL of PEG-2Mal-2NAC solution was added dropwise to the Alg-SH-PIM solution (Table 2) with vortexing, and the mixture was agitated at 25 °C for 1 h to form the Alg-PPN(Cn) solution. The Alg-PPN(Cn) solution was loaded into a syringe and extruded through a 25 G needle into a 250 mL bath of 0.1 g/mL aqueous CaCl$_2$ solution. The formed fibers were passed along the CaCl$_2$ bath and pulled into an ethanol bath for dehydration. The fibers were sonicated in DI water for 1 h and immersed in DI water for 24 h to wash out unreacted components. After washing, the fibers were dehydrated in ethanol and dried under ambient air to obtain Alg-PPN(Cn) fibers.

## Swelling dynamics and mechanical characteristics of the film and fiber hydrogels

Prior to testing of swelling dynamics, hydrogels were thoroughly washed with DI water and then dried with a freeze dryer. The mass of a fully dried hydrogel was measured as the initial mass. The fully dried gel was then immersed in a copious amount of DI water to swell. At 5 min intervals, the hydrogel was removed from the DI water bath, dried by blotting on filter paper, weighed, and then returned to the DI water bath to continue swelling. The swelling ratio was calculated as:

$$\text{Swelling ratio} = \frac{\text{mass of hydrogel at n}^{\text{th}}\text{ min} - \text{initial mass of hydrogel}}{\text{initial mass of hydrogel}} \quad (1)$$

The mechanical strength of the hydrogel was evaluated by performing uniaxial tensile tests using an MTS Criterion 43 (Instron Materials Test system).

## Degradability test of the PPN(C4)−1 film hydrogel in bacterial extracts

Bacterial extracts were prepared by shaking various concentrations of bacteria in PBS at 37 °C and pH 7.4 for 24 h, followed by centrifugation to separate the bacteria from the soluble extracts. PPN(C4)-1 hydrogels were equilibrated by immersion in PBS for 24 h in a 24-well plate, after which the initial masses were weighed. The hydrogels were then incubated in 1 mL of bacterial extract at 37 °C for 2 or 7 days, after which the gels were weighed.

## Degradability test of the PPN(C4)-1 film hydrogel in wound fluids and in infected animal wounds

For the test of stability in wound fluids, mouse wound tissue infected with MRSA USA300 or CR-PA was homogenized in PBS (900 μL), and diluted to a total volume of 10 mL in PBS. After dilution, tissue debris was removed by centrifugation. PPN(C4)-1 hydrogels were equilibrated by immersion in PBS for 24 h in a 24-well plate, after which initial gel masses were measured. Equilibrated hydrogels were then incubated in 1 mL of wound fluid at 37 °C for 2 or 7 days, after which they were weighed. Hydrogel stability on infected wounds was

assessed by photographing PPN(C4)-1 hydrogels before and after application (for 2 days) to treat mouse wounds infected with MRSA USA300.

## Determination of the thiol group content of Alg-SH

The thiol group content of Alg-SH was measured using reaction with Ellman's reagent (DTNB)[57]. Alg-SH (1.76 mg, 10 μmol) was dissolved in 0.1 mL of PBS at neutral pH. DTNB (4 mg) was dissolved in 0.1 mL PBS at neutral pH. The Alg-SH solution (0.1 mL) and DTNB solution (0.1 mL) were mixed in 1.8 mL of PBS and agitated at 100 rpm for 20 min in the dark. Then, the absorbance at 412 nm was measured on a UV–Vis spectrophotometer (Shimadzu UV-1800) and used to determine the thiol group concentration. A calibration curve was prepared by reacting DTNB with L-cysteine at predetermined concentrations. The thiol content determination was validated with fluorescence analysis via a click reaction with F5M. The Alg-SH (3.52 mg) was dissolved in 0.5 mL of DI water. Separately, F5M (4.27 mg, 10 μmol) was dissolved in 0.5 mL of DMSO. The Alg-SH solution and F5M solution were mixed, agitated at 100 rpm for 24 h, and dialyzed (MWCO 1 kDa) for 3 days in the dark. After lyophilization, a solution of Alg-F5M (1.76 mg/mL) was prepared and the fluorescence intensity was measured on a Shimadzu RF-6000 spectrofluorophotometer under an excitation/emission wavelength of 494/518 nm. A calibration curve was prepared by measuring the fluorescence intensity of F5M at predetermined concentrations.

## Determination of the leaching rates of components from the Alg-PPN fibers

The Alg-PPN(C4) fibers (100 mg) were immersed in 2 mL of PBS (for absorbance test) or ultrapure water (for mass spectroscopy test) at 37 °C and agitated at 100 rpm for 24 h. Then, 1 mL of the supernatant was collected. The extracted ion chromatogram (EIC) of the supernatant was obtained on a liquid chromatography-mass spectrometer (Agilent 6550 iFunnel Q-TOF) with Acquity UPLC HSS T3 column (Waters). The absorbance of the supernatant was measured on a UV-Vis spectrophotometer (Shimadzu UV-1800). PEG-4Mal exhibited absorbance peak at 300 nm wavelength. Calibration curves of the components were prepared by measuring the EIC and spectral absorbance of PIM(C4), NAC, and PEG-4Mal at predetermined concentrations.

## Minimum inhibitory concentration (MIC) assay

Bacteria were cultured in MHB at 37 °C with continuous shaking at 220 rpm to mid log phase. Two-fold serial dilutions (1024 μg/mL to 2 μg/mL) of PIM(C4)-Mal in 50 μL MHB were prepared on a 96-well plate. Then, 50 μL bacterial suspension ($2 \times 10^5$ CFU/mL) was added to each well containing PIM(C4)-Mal solutions. The plate was then incubated at 37 °C for 18 h, and the optical density of the wells was measured at 600 nm wavelength to determine the MIC.

## Minimum biofilm eradication concentration (MBEC) assay

The MBEC was measured using a microtiter plate-based technique. Briefly, 180 μL of MRSA LAC or PAO1 suspension (cell density at ~$10^7$ CFU/mL) in TSB was added to a 96-well plate covered by a lid containing pegs (Innovotech 19111). Biofilms were grown on the peg lid after incubation at 37 °C for 24–48 h. After removing planktonic bacteria by washing twice with PBS, the lid with biofilms was transferred into a 96-well challenge plate containing two-fold serial dilutions of polymer solutions with a total volume of 200 μL in each well. The treatment was performed at 37 °C for 4 h. After that, the peg lid was washed with PBS and transferred to a 96-well recovery plate containing 200 μL of neutralizer (3% lecithin, 10% tween 80, and 0.3% sodium thiosulfate in DI water) in each well. Biofilm bacteria were dislodged from the peg lid by sonication (150 kW, 37 kHz) for 30 min. The detached bacteria were then serially diluted 10-fold in PBS and spread on LB agar plates. After incubation at 37 °C for 24 h, bacterial colonies were counted.

## Ex vivo 3D de-epidermised dermis human skin equivalent (DED-HSE) model

**Cell culture.** Primary human keratinocytes were obtained from Asian Skin Bank, A*STAR, with ethical approval IRB: B-16-135E. Cells were cultured in full growth (FG) medium with irradiated 3T3 fibroblasts (i3T3) as feeder cells[58,59]. Briefly, $1 \times 10^6$ i3T3 feeder cells were pre-seeded in a Petri dish of 10 cm diameter. The feeder cells were cultured in Dulbecco's Modified Eagle's Medium (DMEM, Life Technologies, Singapore) containing 10% fetal calf serum (FCS, Life Technologies, Singapore) and 1% v/v penicillin/streptomycin solution at 37 °C with 5% $CO_2$/95% air. After the attachment of i3T3, which requires a minimum of 2 h, 0.5 million primary human keratinocytes were seeded in each Petri dish with the i3T3, and the DMEM was replaced with fresh FG medium. The cells were maintained in an incubator at 37 °C with 5% $CO_2$/95% air, and the medium was replaced every 2–3 days.

**Construction of the DED-HSE wound healing model.** The DED-HSE wound healing model was established using decellularized dermis derived from human skin[59]. Briefly, large pieces of human skin were trimmed into 1 cm$^2$ pieces and soaked in 1 M NaCl overnight, yielding decellularized dermis. The epidermal layer was then removed, and sterile stainless-steel rings were placed onto the papillary side of each de-epidermised dermis (DED). Keratinocytes ($2 \times 10^4$) were transferred into each ring placed on the DEDs and incubated with 5% $CO_2$/95% air at 37 °C for 2 days to permit attachment. The reconstructed samples were then transferred to an air-liquid interface and incubated for another 9 days to allow expansion of the epidermal layer. A 4-mm diameter superficial excision wound was created in the DED-HSEs using a 4-mm biopsy punch (Integra Miltex, Fisher Scientific), and the reformed epidermal layer was removed. Then, sterile topical treatments were performed by adding the hydrogels directly to the wound bed surface for 4 days and 7 days.

**MTT staining.** The lateral migration of viable keratinocytes was observed using the MTT assay. Viable cells with active metabolism react with the tetrazolium ring of the MTT reagent to produce a purple formazan product. A frozen aliquot of a 10-fold volume of MTT stock of 5 mg/mL (Sigma M5655) was diluted 1:10 in PBS. The DED-HSE samples were placed in a 24-well plate, and each well contained 1 ml of MTT reagent. The samples were incubated in a 5% $CO_2$/95% air incubator at 37 °C for 90 min to allow the formation of formazan product. Images of the DED-HSEs were captured using a Nikon SMZ745T microscope. Quantification of the unhealed wound area was performed using ImageJ software.

**Hematoxylin and eosin (H&E) staining.** On days 4 and 7 after wounding, the DED-HSEs following MTT analysis were fixed in formalin overnight before embedding in paraffin. Sections of 7 μm thickness were cut and transferred onto glass slides. These sections were dewaxed in xylene and rehydrated in descending concentrations of ethanol before H&E staining. Images were captured using an Olympus BX43 microscope.

**Immunohistochemistry (IHC).** DED-HSEs were fixed, embedded, cut into thin sections, dewaxed and rehydrated as described above. Sections were then immersed in an antigen retrieval, citrate buffer pH 6.0, and heat-induced epitope retrieval was performed in a 90 °C water bath for 15 min. 1% $H_2O_2$ was used to quench endogenous peroxidase for 30 min. Then, the sections were blocked and incubated with anti-K10 (1:200), anti-K14 (1:25), and anti-p63 (1:50) primary antibodies for 2 h at room temperature. The slides were washed with PBS containing 0.05% Tween 20 and incubated with horseradish peroxidase (HRP)-labeled anti-mouse polymer (1:200) and HRP-labeled anti-rabbit polymer (1:200) secondary antibodies for 1 h at room temperature. IHC sections were developed with diaminobenzidine (DAB) substrate,

and nuclei were counterstained with hematoxylin, dehydrated through ascending concentrations of ethanol to xylene, and finally mounted onto coverslips. Images were captured using an Olympus BX43 microscope.

## Antimicrobial properties of the hydrogel films and fibers in vitro

**Antimicrobial assay towards planktonic bacteria.** Bacteria (MRSA USA300, CR-AB, PAO1 or CR-PA) were inoculated into MHB (4 mL) and cultured at 37 °C with shaking at 220 rpm to a mid-log phase. Cultured bacteria were collected by centrifugation and decanting of the supernatant. Collected bacteria were washed thrice with PBS and suspended in PBS at a concentration of $10^9$ CFU/mL. Hydrogel film or fiber samples were inoculated with 10 μL of bacterial suspension (containing $10^7$ CFU) spread evenly on the sample surface. A Petri dish without hydrogel film or fiber sample was inoculated as a control. The samples were incubated at 37 °C and 90% relative humidity for 1 h. After incubation, bacteria were released from the samples by immersion in 1 mL of PBS and agitation with a vortex mixer. The obtained bacterial suspensions were serially diluted 10-fold with PBS in a 96-well plate, and plated on LB agar. The agar plates were incubated at 37 °C for 16 h, and the bacterial colonies were counted. The results are reported as:

$$\text{Log reduction} = \text{Log(total CFU of control)} - \text{Log(total CFU on hydrogels)} \tag{2}$$

**Biofilm contact killing assay.** Sterile 1 cm × 1 cm sections of 0.2-μm polycarbonate membrane (Whatman, New Jersey) were placed on LB agar plates. A bacterial suspension in PBS (10 μL) containing approximately $10^5$ CFU was inoculated onto the membrane surface. The membranes were incubated at 37 °C for 24 h to form biofilm gels. The biofilms on the membrane were then treated by contact with fiber samples and incubated at 37 °C for 24 h. Bacteria were liberated by vortexing the membrane in Dey/Engley medium, serially diluted 10-fold in PBS, and then plated on LB agar. The LB agar plates were incubated at 37 °C for 24 h, and the bacterial colonies were counted.

## In vitro biocompatibility assay of PIM-Mal and hydrogels

Biocompatibility studies were carried out on mouse 3T3 fibroblasts and HDFs (NHDF-Ad-Der Fibroblasts, CC2511, Lonza). DMEM fully supplemented with fetal bovine serum (FBS, 10%), L-glutamine (1 mM), and antibiotics (penicillin–streptomycin, 1%) was used as cell culture medium.

**MTT assay of the hydrogel extracts.** Hydrogel film or fiber samples were placed in each well of a 24-well plate with 1 mL of DMEM and incubated at 37 °C for 24 h to collect the extracts. HDF or 3T3 cells were cultured in 1 mL DMEM in 24-well plates from an initial density of $5 \times 10^4$ cells in each well and incubated in a 5% $CO_2$ incubator at 37 °C for 24 h for cell attachment. The DMEM was aspirated and replaced with the sample extracts (or fresh DMEM for positive control) to incubate with the cells at 37 °C for 24 h. Then, the culture media were replaced with 1 mL MTT solution (0.5 mg/mL in DMEM) and incubated at 37 °C for 4 h to stain viable cells. The MTT solution was gently aspirated, and 1 mL dimethyl sulfoxide (DMSO) was added to each well. After 15 min, the absorbance of each well at 570 nm was measured using Tecan i-control microplate reader. The absorbance of DMSO was measured as a blank control. The cell viability was calculated using the equation:

$$\text{Relative cell viability(\%)} = \frac{A_S - A_B}{A_P - A_B} \times 100\% \tag{3}$$

where $A_S$, $A_B$, and $A_P$ are the absorbance of the sample, blank, and positive control, respectively.

**Direct contact MTT assay.** The procedures were the same as above (MTT assay of the hydrogel extracts), but instead of immersing the cells with supernatant extracts, the hydrogel film or fiber samples were directly immersed in the culture medium in the wells containing the attached cells for 24 h and then removed before replacing the medium with MTT solution.

## In vivo murine model of infected diabetic wound treatment

All mouse studies were carried out under the regulation of the Institutional Animal Care and Use Committee (IACUC) of Nanyang Technological University (NTU) under the approved protocol numbers A18051 and A21023.

**Diabetic induction on mice.** Eight-week-old male C57BL/6 mice were conditioned by fasting for 4 h prior to the start of diabetic induction injections. Diabetes was induced with streptozotocin (STZ) solution, 4 mg STZ dissolved in 1 mL of sodium citrate buffer (50 mM) at pH 4, injected intraperitoneally at dosage 40 mg/kg daily for 5 consecutive days. Liver Disease Progression Aggravation Diet (LIDPAD, provided by N.S. Tan's lab (NTU)) and 10% sucrose water solution were supplied during the 5 days of injection and then changed to regular diet and water on day 6 onward. Mouse blood glucose levels were measured 3 weeks after the end of injection. The mice were assessed to be diabetic if their blood glucose level surpassed 11.1 mmol/L.

**Bacterial load enumeration on skin wound infection model.** Mice were anesthetized and depilated, and a 6-mm diameter full-thickness excisional wound was created on the dorsal skin and the underlying panniculus carnosus[60]. In total, 10 μL of bacterial suspension (MRSA USA300, CR-AB, PA01 or CR-PA at $10^6$ CFU in PBS) was inoculated onto the wounds and left to settle for 10 min before covering the infected wounds with transparent Tegaderm patch (3 M). The wounds were left untreated for 24 h to allow the inoculated bacteria to form biofilms. Hydrogel film or fiber dressings were applied as treatment condition after the 24 h infection period. Untreated wounds served as controls. After 24 h, the dressings were removed and the wounds (including 5 mm of the peripheral region) were excised. For biofilm imaging, excised wound tissue was placed in PBS (1 mL) and observed with an LSM 800 confocal microscope (ZEISS). For bacterial counting, excised wound tissue was homogenized in 900 μL of PBS to release the bacteria ($n = 6$). The bacterial suspension was serially diluted 10-fold with PBS in a 96-well plate. The diluted suspensions were plated on LB agar, and the agar plates were incubated at 37 °C for 16 h, after which the bacterial colonies were counted. The two-tailed Student's $t$-test was used for statistical analysis.

**Wound healing study.** As described above, wounds infected with MRSA USA300 or CR-PA biofilms were created on mice. At day 0, untreated wounds were secured with Tegaderm as control; hydrogel film or fiber dressing, secured with Tegaderm, was applied to treatment condition wounds. The wounds were photographed before dressing application and on days 1, 3, 5, 7, 9, 12 and 14 of the treatment period, at which time points the dressings were replaced with fresh ones. For Alg-PPN treatment group, Alg-PPN(C8)−5 was applied to the wound until day 5 to eradicate bacteria, followed by replacement with Alg-PPN(C4)-5 from day 5 to day 14. At each time point, the wound size was measured using ImageJ software ($n = 6$). The two-tailed Student's $t$-test was used for statistical analysis. Relative wound size is reported as:

$$\text{Wound size(\%)} = \frac{\text{wound area on } n^{\text{th}} \text{day}}{\text{wound area on day 0}} \times 100\% \tag{4}$$

**Fluorescence-activated cell sorting (FACS) analysis of inflammatory cells.** Wounds (including 5 mm of the peripheral region) were excised after 2 days of treatment of the mouse wound

healing study. Single-cell suspensions were obtained from the wound samples using a gentleMACS Dissociator (Miltenyi Biotec). The cells were immune-labeled with CD11b$^+$ and Ly6G, and cytometry was performed using an Accuri C6 flow cytometer (BD Biosciences). The cytometry data was analyzed using FlowJo software (version 7.6.5, Tree Star). Analysis results are plotted as mean percentage values ± standard error of the mean (SEM, $n = 6$). The two-tailed Student's $t$-test was used for statistical analysis.

**Enzyme-linked immunosorbent assay (ELISA) for wound healing-related factors.** After 2 days of treatment of the mouse wound healing study, wounds (including 5 mm of the peripheral region) were excised, homogenized in 900 μL of PBS and centrifuged to remove tissue and bacteria. The supernatants were assayed with various ELISA kits (pro-MMP9, VEGF-A, PDGF-BB, FGF-2 and EGF) according to the manufacturer's protocols (Lonza). The results for each treatment are plotted as mean concentration values ± SEM ($n = 6$). The two-tailed Student's $t$-test was used for statistical analysis.

### Histological analysis and histochemical staining
On day 7 post-treatments, animals were sacrificed using $CO_2$ asphyxiation. The wound tissue, along with the surrounding tissue, was carefully excised and fixed in 10% formalin. Subsequently, tissue samples were dehydrated, embedded in paraffin wax blocks, and cut into 5 μm thick sections using a Leica Microtome (Leica Biosystems). These sections were then collected on Superfrost Plus glass slides (ThermoFisher Scientific). The sections were processed and stained with hematoxylin and eosin (H&E) and picrosirius red. The stained slides were imaged using Axio Scan.Z1 microscope system (ZEISS). Granulation tissue thickness and collagen content were quantified using ImageJ software.

### Reporting summary
Further information on research design is available in the Nature Portfolio Reporting Summary linked to this article.

## Data availability
The experimental data generated in this study are provided within the Article and Supplementary Information. All data are available from the corresponding authors upon request.

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

## Acknowledgements

We thank Dr. Xi Liu (Institute of Microbiology of the Chinese Academy of Sciences) and Dr. Ke Wang (Guangxi Medical University) for their advice in the animal study. This work was funded and supported by an A*STAR Industry Alignment Fund Pre-Positioning Programme (IAF-PP, HBMS Domain) H17/01/a0/0B9, H17/01/a0/0M9, and H19/01/a0/OY9 as part of the Wound Care Innovation for the Tropics (WCIT, HBMS Domain, H17/01/a0/009) and the Singapore MOE Tier 3 grant (MOE2018-T3-1-003). This research was also supported by the National Research Foundation, Prime Minister's Office, Singapore, under its Campus for Research Excellence and Technological Enterprise (CREATE) programme, through the SMART AMR IRG. C.F, P.L.L.K. and D.I.L. acknowledge infrastructure support from A*STAR (IAF-PP, HBMS Domain, H17/01/a0/004). C.K.Y. acknowledges the support of NTU IGS-HealthTech Ph.D. scholarship and the A*STAR IAF (WCIT, H17/01/a0/009).

## Author contributions

D.P. performed the experimental work of film hydrogel and alginate fiber hydrogel. C.K.Y. performed the experimental work of film hydrogel. D.P. and C.K.Y. synthesized the polyimidazolium and hydrogel dressings, and performed the in vitro and in vivo experiments. Y.W., C.F., P.L.K.L., and D.I.L. designed, performed, and interpreted the ex vivo human skin equivalent wound model experiments. X.X. performed the antibiofilm assay of polyimidazolium. Y.S.Y. and M.I.G.V. assisted in the mouse experiment. S.H.M. established the chemistry of polyimidazolium syntheses and performed NMR characterization. N.S.T. and L.Y. advised on the design and interpretation of in vivo diabetic mouse wound model experiments. M.B.C.-P. advised on the design and interpretation of all experiments and directed the overall project. D.P., C.K.Y. and M.B.C.-P. wrote and edited the manuscript. P.T.H., D.I.L. and N.S.T. edited the manuscript. D.I.L. and M.B.C.-P. acquired the funding to support this project.

## Competing interests

D.P., C.K.Y., Y.W., X.X., N.S.T., and M.B.C.-P. have filed the Singapore provisional patents for this work under the patent numbers

10202250243U and 10202301624S. C.K.Y., S.H.M., N.S.T., and M.B.C.-P. have filed a patent for this work under the patent number US2021O244846A1. D.P., C.K.Y, Y.W., X.X., S.H.M, N.S.T, and M.B.C.-P. declare no other competing interests. C.F., Y.S.Y., M.I.G.V., P.L.K.L., L.Y., P.T.H., and D.I.L. declare no competing interests.
