## [Peer Review File · Nature Communications]

Hydrogel dressings with intrinsic antibiofilm and antioxidative dual functionalities accelerate infected diabetic wound healingReviewers' comments:

Reviewer #1 (Remarks to the Author):

In this report, the authors reported a crosslinked PEG hydrogel containing antibacterial cationic polyimidazolium and antioxidant compound N-acetylcysteine (NAC). The hydrogels have been investigated for anti-bacterial and wound healing applications. Overall, using functional PEG hydrogels for wound healing and anti-bacterial have been reported a lot and this manuscript did not show reasonable novelty or significance. The performance of this hydrogel is not good, especially for wound healing applications. Therefore, this manuscript may not be suitable for Nature Communications.

Major issues:

- 1, the experimental design is weak, the hydrogels need many extra characterizations, such as swelling mechanics, degradation behavior, mechanical properties, hydrogel micro/nano morphology
- 2, the authors need more control groups and the current control group selection is not very reasonable, more commercial wound healing dressing are needed, or ointments
- 3, the results are not very encouraging in terms of wound healing, for pig model, only at the last time point, there is some difference
- 4, more mechanism studies are needed for why it could anti-bacterial and wound healing

Minor issues:

- 1, the title is not suitable, many hydrogels are removable hydrogel and the authors did not show the details how good it is in terms of removable capability
- 2, there are a great number of multi-arm based PEG hydrogels, what's the novelty of this system?
- 3, writing quality need to be improved
- 4, some unimportant results could be moved to SI, such as Figure 6

Reviewer #2 (Remarks to the Author):

General Comments: In this paper, the authors have summarized the application of polyethylene glycol (PEG) hydrogels with alginate fibers containing polyimidazolium as a broad-spectrum antibiotic against biofilm forming multidrug resistant Gram-positive and Gram-negative bacteria, and N-acetylcysteine as an antioxidant which promotes epithelialization. The paper includes characterization of the hydrogels and treatment of 3D De-Epidermised Dermis Human Skin Equivalent (DED-HSE) to study wound closure, re-epithelialization, and proliferation and differentiation of human keratinocytes, as well as an infected diabetic murine model for flat wounds, and an infected porcine model for deep wounds.

Overall, the focus of this manuscript is based on the development of PEG hydrogels and alginate fibers

containing polyimidazolium as a broad-spectrum antibiotic, and N-acetylcysteine as an antioxidant to enhance wound closure. Both in vitro and multiple in vivo models are utilized, including an infected diabetic murine model and a porcine model. The inclusion of the human ex vivo model is noteworthy. The authors show significant reduction of bacterial load from infected wounds in the mouse model, but not complete eradication. It unclear the benefit in the pig model beyond slightly faster closure. Overall, the main finding is improved wound closure rates with PPN(C4)-1 treatment, and wounds showed progress towards healing via expression of wound healing factors.

Comments:

1. In multiple places the manuscript the authors claim that a biofilm is formed and treated, however, characterization of the biofilm and effects of treatments is not included in the paper. Namely, there would need to be analysis of bacterial density, film organization and of expression of extracellular molecules. The statement 'Pus and sluff were observed on the untreated control wounds' is insufficient. Going on to say the treatments have 'antibiofilm effects' is uncertain without proper analyses of the biofilm structure.
2. Figure 2A: how many replicates were completed for bacterial quantification? No error bars are presented. Biological replicates for bacterial counts should always be a minimum of triplicate.
3. Instead of only including histology for the DED-HSE tissue, it would have been useful to include histology staining of the diabetic murine model, as wound healing in a diabetic system is delayed as compared to a non-diabetic system. Additionally, the murine model does not seem to be splinted, which could have been a way to model healing via re-epithelialization rather than via contraction, although we recognize the utilization of DED-HSE and porcine models to show re-epithelialization.
4. Figure 4B: there is some concern that full eradication of MRSA cannot be completed within 14 days of treatment using PPN(C4)-1. The sustained population of MRSA at D14 even with PPN(C4)-1 treatment has potential to recolonize the wound with removal of the treatment.
5. The focus of this paper is on infection control and wound closure in terms of both animal models. Based on this observation the models are problematic as the infected controls will heal on their own, it appears. This is a common problem in the field, as is only having a single endpoint for main analyses. The authors of this study picked an endpoint that makes sure the main treatment is different from the control in terms of reepithelization. The question is what if a later time point was selected for when the control also closes (e.g., at 21, 28, 35, etc. days)? Would the treatment matter then for the metrics and analyses selected? This is an important question as it's possible a complex treatment isn't needed, only simply allowing more time for the body to fight the infection in these animal models.
6. Figure 4B: Clarify in the text if all wound counts were collected in the same way as in 4A, with animals sacrificed and full wounds excised and homogenized on each day of bacterial quantification.
7. The pig study is problematic and incomplete. Namely, histological and bacterial analyses are missing as were done in the other studies (ex vivo and mouse). The closure data shows very minor, albeit significant, differences from the authors' treatment. In Figure 8b, the treatment (Alg-PPN(C8)-5/Alg-

PPN(C4)-5) also exhibits a dark greenish/purple film which is not seen in other treatments at the later timepoints, is this partially bacterial byproducts? I see this in the supplemental data as well. I also see a mix of before and after debrided pictures for each timepoint, why?

Reviewer #3 (Remarks to the Author):

The paper “Removable hydrogel dressings with antibiofilm and antioxidation dual 2 functionalities accelerate infected diabetic wound healing” proposes a new type of functional hydrogel dressing (PPN) with antibiofilm and antioxidant properties by combining a crosslinked PEG hydrogel with covalently linked antibacterial cationic polyimidazolium and N-acetylcysteine (NAC), which possesses antioxidant properties and good integrity over time. This paper makes its case by demonstrating that this PPN hydrogel is able to fairly accelerate the closure of wounds infected with antibiotic-resistant bacteria either in a murine diabetic wound model or when combined in an alginate fiber, in a pig wound model.

The versatility of the hydrogel is an interesting output from this work as the method for its production allows for different concentrations of the active components (PIM(Cn)-Mal and NAC) to be grafted, to treat different severities/stages of wounds.

The authors claim their major novelty to be the targeting of infection on non-healing wounds. This is an impactful concept, however the same rationale has been recently proposed in the literature, and this has not been fully covered in the discussion, such as for example the following publications:

- Shiekh PA, Singh A, Kumar A. Exosome laden oxygen releasing antioxidant and antibacterial cryogel wound dressing OxOBand alleviate diabetic and infectious wound healing. *Biomaterials*. 2020 Aug 1;249:120020.

- Ma T, Zhai X, Huang Y, Zhang M, Zhao X, Du Y, Yan C. A smart nanoplatfrom with photothermal antibacterial capability and antioxidant activity for chronic wound healing. *Advanced Healthcare Materials*. 2021 Jul;10(13):2100033.

- Ge P, Chang S, Wang T, Zhao Q, Wang G, He B. An antioxidant and antibacterial polydopamine-modified thermo-sensitive hydrogel dressing for *Staphylococcus aureus*-infected wound healing. *Nanoscale*. 2022.

The authors should present the benefits of the presented hydrogels over existing systems.

Other recent work proposing the dual effect of antibacterial and antioxidant effect have been published and has not been duly addressed:

- Liang Y, Zhao X, Hu T, Han Y, Guo B. Mussel-inspired, antibacterial, conductive, antioxidant, injectable

composite hydrogel wound dressing to promote the regeneration of infected skin. *Journal of colloid and interface science*. 2019 Nov 15;556:514-28.

Regarding the results, in general the presented work gathers an extensive characterization which demonstrates well the physicochemical properties of the materials and the biofunctionality of the proposed hydrogel and alginate-based fibers while promoting some healing efficacy as wound dressings as compared with the controls, in different *in vitro* and *in vivo* models.

However, more important than the healing rate is the quality of the regenerated tissue. The characterization of the collagen quality should be assessed to understand the type of new tissue that is being built, besides the information already provided in the histological characterization and presence of wound healing factors.

The hydrogel has very interesting properties and performance, mostly the fact that it has a fast crosslink, and it has good integrity which allows to be periodically changed. However there are no clear evidences on the mechanical properties during the removal of the hydrogel after being in contact with the wound. Does it leave residues? Macroscopic images or a video would be helpful to fully demonstrate its integrity. Moreover, the swelling kinetics of hydrogels are not fully discussed in the paper. It is important to better explore the fluid managing capacity of these materials (hydrogel and fibers) over time, according with the type of wound. Water retention and water-vapour permeability is another important property that needs to be assessed to help define the specific type of wound to be addressed. The discussion should clearly bring the publications mentioned above into light.

For the most part, the work is technically acceptable, but lacks some i

The paper "Removable hydrogel dressings with antibiofilm and antioxidation dual functionalities accelerate infected diabetic wound healing" proposes a new type of functional hydrogel dressing (PPN) with antibiofilm and antioxidant properties by combining a crosslinked PEG hydrogel with covalently linked antibacterial cationic polyimidazolium and N-acetylcysteine (NAC), which possesses antioxidant properties and good integrity over time. This paper makes its case by demonstrating that this PPN hydrogel is able to fairly accelerate the closure of wounds infected with antibiotic-resistant bacteria either in a murine diabetic wound model or when combined in an alginate fiber, in a pig wound model.

The versatility of the hydrogel is an interesting output from this work as the method for its production allows for different concentrations of the active components (PIM(Cn)-Mal and NAC) to be grafted, to treat different severities/stages of wounds.

The authors claim their major novelty to be the targeting of infection on non-healing wounds. This is an impactful concept, however the same rationale has been recently proposed in the literature, and this has not been fully covered in the discussion, such as for example the following publications:

- Shiekh PA, Singh A, Kumar A. Exosome laden oxygen releasing antioxidant and antibacterial cryogel wound dressing OxOBand alleviate diabetic and infectious wound healing. *Biomaterials*. 2020 Aug 1;249:120020.

- Ma T, Zhai X, Huang Y, Zhang M, Zhao X, Du Y, Yan C. A smart nanoplatfom with photothermal

antibacterial capability and antioxidant activity for chronic wound healing. *Advanced Healthcare Materials*. 2021 Jul;10(13):2100033.

- Ge P, Chang S, Wang T, Zhao Q, Wang G, He B. An antioxidant and antibacterial polydopamine-modified thermo-sensitive hydrogel dressing for *Staphylococcus aureus*-infected wound healing. *Nanoscale*. 2022.

The authors should present the benefits of the presented hydrogels over existing systems.

Other recent work proposing the dual effect of antibacterial and antioxidant effect have been published and has not been duly addressed:

- Liang Y, Zhao X, Hu T, Han Y, Guo B. Mussel-inspired, antibacterial, conductive, antioxidant, injectable composite hydrogel wound dressing to promote the regeneration of infected skin. *Journal of colloid and interface science*. 2019 Nov 15;556:514-28.

Regarding the results, in general the presented work gathers an extensive characterization which demonstrates well the physicochemical properties of the materials and the biofunctionality of the proposed hydrogel and alginate-based fibers while promoting some healing efficacy as wound dressings as compared with the controls, in different *in vitro* and *in vivo* models.

However, more important than the healing rate is the quality of the regenerated tissue. The characterization of the collagen quality should be assessed to understand the type of new tissue that is being built, besides the information already provided in the histological characterization and presence of wound healing factors.

The hydrogel has very interesting properties and performance, mostly the fact that it has a fast crosslink, and it has good integrity which allows to be periodically changed. However there are no clear evidences on the mechanical properties during the removal of the hydrogel after being in contact with the wound. Does it leave residues? Macroscopic images or a video would be helpful to full demonstrate its integrity. Moreover, the swelling kinetics of hydrogels are not fully discussed in the paper. It is important to better explore the fluid managing capacity of these materials (hydrogel and fibers) over time, according with the type of wound. Water retention and water-vapour permeability is another important property that needs to be assessed to help define the specific type of wound to be addressed. The discussion should clearly bring the publications mentioned above into light.

For the most part, the work is technically acceptable, but lacks some useful information and a deep interpretation or discussion

Replies to the Reviewers' Comments

Reviewer #1 (Remarks to the Author):

In this report, the authors reported a crosslinked PEG hydrogel containing antibacterial cationic polyimidazolium and antioxidant compound N-acetylcysteine (NAC). The hydrogels have been investigated for anti-bacterial and wound healing applications. Overall, using functional PEG hydrogels for wound healing and anti-bacterial have been reported a lot and this manuscript did not show reasonable novelty or significance.

Reply: This paper is not about PEG-hydrogel but about a dual functionality wound dressing device of antibacterial and antioxidant hydrogel that is non-leachable. The active components are a contact-active cationic polyimidazolium (PIM) polymer that is highly potent and an antioxidant NAC that quenches diffusible ROS and both these components are non-leachable. The components of the wound dressing are non-leachable and would be classified by FDA as a device rather than a drug as many reports drug like wound dressing. According to the FDA classification, our PPN hydrogel should fall under Class II (21 CFR 878.4015): a wound dressing type where a high-charge density cationic polymer with antimicrobial activity (such as poly(diallyl dimethyl ammonium chloride)) is permanently bound to the substrate (such as textile).

These clarifications are now mentioned in Introduction (on Page 6, line 1-6) and in Discussion (on Page 24, second paragraph, 1st – 5th sentences).

The performance of this hydrogel is not good, especially for wound healing applications. Therefore, this manuscript may not be suitable for Nature Communications.

Reply: The wound healing is much improved and faster compared with current golden standard of Silver hydrogel. Please see the Figure 4 and Figure 6, and the positive comments of Reviewers 2 and 3 too stating this point. Maybe the reviewer has not read the manuscript in details.

We have mentioned that the PPN hydrogel removed more bacteria and facilitated faster healing compared to silver dressing (on Page 13, second paragraph, 3rd – 7th sentences). PPN hydrogel also reduced inflammatory cells (on Page 14, first paragraph, 3rd – 4th sentences) and elevated the level growth factors in wound healing compared to silver dressing (on Page 14, second paragraph, 4th sentence).

Major issues:

1, the experimental design is weak, the hydrogels need many extra characterizations, such as swelling mechanics, degradation behavior, mechanical properties, hydrogel micro/nano morphology

Reply: I do not think the experimental design is weak because the paper already has 55 pages and 6 figures (plus additional 16 supplementary figures and 5 supplementary extended tables). But we will get the data/parameters that the reviewer asked, which can be supplemented by additional experiments during the revision (including swelling mechanics, degradation behaviour, mechanical properties, and hydrogel morphology).

2, the authors need more control groups and the current control group selection is not very reasonable, more commercial wound healing dressing are needed, or ointments

Reply: I do not think ointment is suitable as a control because ointment is not removable and contaminates the wound and infiltrates into the body. Our hydrogel is about a non-intrusive hydrogel

dressing that is removable. The silver dressing is the golden standard in the market and we had chosen 2 controls – silver hydrogel and silver alginate fiber.

3, the results are not very encouraging in terms of wound healing, for pig model, only at the last time point, there is some difference

Reply: I agree the pigs' test was interrupted because of COVID in China, and we cannot easily repeat then because of lockdown. I wish to remove the pigs' data, if you and the other reviewers agree since the paper is already quite illustrative of our novelty of dual functionality wound dressing with in vitro, 3D model and mice studies.

However, if pigs' studies must be inside, we can try to repeat this again in China since it is now open.

4, more mechanism studies are needed for why it could anti-bacterial and wound healing

Reply: The mechanism has been studied in details and explained in Figures 4 and 6, which showed that the bacteria have been removed, and the antioxidant helps reduce the ROS.

Minor issues:

1, the title is not suitable, many hydrogels are removable hydrogel and the authors did not show the details how good it is in terms of removable capability

Reply: We now deleted the word "removable" from the title. In addition, the PPN hydrogel film and fiber can be removed from wounds due to their sturdiness and robust mechanical integrity. We will carry out experiments to show the removal and test the mechanical properties of the dressings.

2, there are a great number of multi-arm based PEG hydrogels, what's the novelty of this system?

Reply: The novelty of this system is wound dressing with covalently bound non-leaching antimicrobial PIM (cationic polymer with superior broad-spectrum antibiofilm activity among other polymers) and covalently bound antioxidant NAC. Together, the antibiofilm and antioxidative components feature synergistic dual functions to accelerate closure of infected wounds. Our PIM here has imidazolium linkers on the main-chain, which imparts strong antibacterial and antibiofilm properties. Four-arm PEG crosslinked via 'click' chemistry was used to form the platform (hydrogel) with strong/robust mechanical integrity. PPN hydrogel is a non-leaching antibacterial and antioxidative wound dressing which is considered as a biomedical device rather than a drug, and would fall under Class II type of wound dressing under FDA classification (21 CFR 878.4015).

These clarifications are now mentioned in Introduction (on Page 6, line 1-6) and in Discussion (on Page 24, second paragraph, 1st – 5th sentences).

3, writing quality need to be improved

Reply: Two native English speakers (one has an extensive background in wound healing study) have checked and provided corrections to the writing quality. We will further check and improve the English quality during the revision.

4, some unimportant results could be moved to SI, such as Figure 6

Reply: We now moved Figure 6 to Figure S14 in SI (Supplementary Information, on page S24).

Reviewer #2 (Remarks to the Author):

General Comments: In this paper, the authors have summarized the application of polyethylene glycol (PEG) hydrogels with alginate fibers containing polyimidazolium as a broad-spectrum antibiotic against biofilm forming multidrug resistant Gram-positive and Gram-negative bacteria, and N-acetylcysteine as an antioxidant which promotes epithelialization. The paper includes characterization of the hydrogels and treatment of 3D De-Epidermised Dermis Human Skin Equivalent (DED-HSE) to study wound closure, re-epithelialization, and proliferation and differentiation of human keratinocytes, as well as an infected diabetic murine model for flat wounds, and an infected porcine model for deep wounds.

Overall, the focus of this manuscript is based on the development of PEG hydrogels and alginate fibers containing polyimidazolium as a broad-spectrum antibiotic, and N-acetylcysteine as an antioxidant to enhance wound closure. Both in vitro and multiple in vivo models are utilized, including an infected diabetic murine model and a porcine model. The inclusion of the human ex vivo model is noteworthy. The authors show significant reduction of bacterial load from infected wounds in the mouse model, but not complete eradication.

Reply: Thanks for reading our paper thoroughly and for the positive comments.

We can test uninfected mice to confirm that the mice do not have bacteria (which we believe is the situation) during the revision. The wound dressing cannot eradicate the bacteria which are not in contact with the wound dressing and which have penetrated into the tissues.

It unclear the benefit in the pig model beyond slightly faster closure.

Reply: We agree the pigs' test was interrupted because of COVID in China, and we cannot easily repeat then because of lockdown. I wish to remove the pigs' data, if you agree since the paper is already quite illustrative of our novelty of dual functionality wound dressing with in vitro, 3D model and mice studies. However, if pigs' studies must be inside, we can try to repeat this again in China since it is now open.

Overall, the main finding is improved wound closure rates with PPN(C4)-1 treatment, and wounds showed progress towards healing via expression of wound healing factors.

Reply: Thanks for your positive comments of appreciation of our novelty.

Comments:

1. In multiple places the manuscript the authors claim that a biofilm is formed and treated, however, characterization of the biofilm and effects of treatments is not included in the paper. Namely, there would need to be analysis of bacterial density, film organization and of expression of extracellular molecules. The statement 'Pus and sluff were observed on the untreated control wounds' is insufficient. Going on to say the treatments have 'antibiofilm effects' is uncertain without proper analyses of the biofilm structure.

Reply: We can do the experiments to characterise the biofilm structure as we did in our previous papers. These biofilm data can be acquired during the revision.

2. Figure 2A: how many replicates were completed for bacterial quantification? No error bars are presented. Biological replicates for bacterial counts should always be a minimum of triplicate.

Reply: We now provided the error bars for bacterial quantification in Figure 2a-d (on Page 50), which were each obtained from three replicates.

3. Instead of only including histology for the DED-HSE tissue, it would have been useful to include histology staining of the diabetic murine model, as wound healing in a diabetic system is delayed as compared to a non-diabetic system.

Reply: We can repeat the histology staining for the diabetic murine model during the revision.

Additionally, the murine model does not seem to be splinted, which could have been a way to model healing via re-epithelialization rather than via contraction, although we recognize the utilization of DED-HSE and porcine models to show re-epithelialization.

Reply: We did not splint the wounds in murine model. However, the control treatment was similarly not splinted. But in the 3D DED-HSE model, we showed the evidence of re-epithelialization, keratinocytes proliferation and differentiation by the PPN hydrogel treatment.

4. Figure 4B: there is some concern that full eradication of MRSA cannot be completed within 14 days of treatment using PPN(C4)-1. The sustained population of MRSA at D14 even with PPN(C4)-1 treatment has potential to recolonize the wound with removal of the treatment.

Reply: The hydrogels can only treat the surface bacteria, and the bacteria that penetrated the tissues cannot be cleared by this wound dressing. It can be re-designed to have some leaching component in the future but the leaching component will penetrate into the body and we need consider the potential toxicity and ease of FDA approval too. We can add on diffusive transient nitric oxide too which we also work on (FDA classified non-leachable wound dressing as devices which have lower hurdle of approval than “drugs” for which wound dressings with leachable components are classified under.)

5. The focus of this paper is on infection control and wound closure in terms of both animal models. Based on this observation the models are problematic as the infected controls will heal on their own, it appears. This is a common problem in the field, as is only having a single endpoint for main analyses. The authors of this study picked an endpoint that makes sure the main treatment is different from the control in terms of reepithelization. The question is what if a later time point was selected for when the control also closes (e.g., at 21, 28, 35, etc. days)? Would the treatment matter then for the metrics and analyses selected? This is an important question as it's possible a complex treatment isn't needed, only simply allowing more time for the body to fight the infection in these animal models.

Reply: We showed that our PPN hydrogel accelerated the closure/healing of infected wounds. With PPN hydrogel treatment, it takes faster time for the wounds to fully heal than the other treatment groups (on Page 13, second paragraph, 5th – 6th sentences, plus Figures 4 and 6). We did not extend the observation days as we have stated the end-point in our animal protocols. We can do other end points too.

6. Figure 4B: Clarify in the text if all wound counts were collected in the same way as in 4A, with animals sacrificed and full wounds excised and homogenized on each day of bacterial quantification.

Reply: Yes, the wound collections in Figure 4b were performed in the same way as in Figure 4a, with animals sacrificed and full wounds excised and homogenized on each day of bacterial quantification. We now clarified this in the main text (on Page 13, second paragraph, 2nd sentence).

7. The pig study is problematic and incomplete. Namely, histological and bacterial analyses are missing as were done in the other studies (ex vivo and mouse). The closure data shows very minor, albeit significant, differences from the authors' treatment. In Figure 8b, the treatment (Alg-PPN(C8)-5/Alg-PPN(C4)-5) also exhibits a dark greenish/purple film which is not seen in other treatments at the later timepoints, is this partially bacterial byproducts? I see this in the supplemental data as well. I also see a mix of before and after debrided pictures for each timepoint, why?

Reply: We agree that the bacterial samples should have been collected but we are unable to easily do so now. We wish to remove the pig data if possible. If not, we can redo it again.

Reviewer #3 (Remarks to the Author):

The paper "Removable hydrogel dressings with antibiofilm and antioxidation dual 2 functionalities accelerate infected diabetic wound healing" proposes a new type of functional hydrogel dressing (PPN) with antibiofilm and antioxidant properties by combining a crosslinked PEG hydrogel with covalently linked antibacterial cationic polyimidazolium and N-acetylcysteine (NAC), which possesses antioxidant properties and good integrity over time. This paper makes its case by demonstrating that this PPN hydrogel is able to fairly accelerate the closure of wounds infected with antibiotic-resistant bacteria either in a murine diabetic wound model or when combined in an alginate fiber, in a pig wound model.

The versatility of the hydrogel is an interesting output from this work as the method for its production allows for different concentrations of the active components (PIM(Cn)-Mal and NAC) to be grafted, to treat different severities/stages of wounds.

Reply: Thanks for reading our paper thoroughly and for the positive comments.

The authors claim their major novelty to be the targeting of infection on non-healing wounds. This is an impactful concept, however the same rational has been recently proposed in the literature, and this has not been fully covered in the discussion, such as for example the following publications:

- Shiekh PA, Singh A, Kumar A. Exosome laden oxygen releasing antioxidant and antibacterial cryogel wound dressing OxOBand alleviate diabetic and infectious wound healing. *Biomaterials*. 2020 Aug 1;249:120020.

- Ma T, Zhai X, Huang Y, Zhang M, Zhao X, Du Y, Yan C. A smart nanoplatform with photothermal antibacterial capability and antioxidant activity for chronic wound healing. *Advanced Healthcare Materials*. 2021 Jul;10(13):2100033.

- Ge P, Chang S, Wang T, Zhao Q, Wang G, He B. An antioxidant and antibacterial polydopamine-modified thermo-sensitive hydrogel dressing for *Staphylococcus aureus*-infected wound healing. *Nanoscale*. 2022.

The authors should present the benefits of the presented hydrogels over existing systems.

Other recent work proposing the dual effect of antibacterial and antioxidant effect have been published and has not been duly addressed:

- Liang Y, Zhao X, Hu T, Han Y, Guo B. Mussel-inspired, antibacterial, conductive, antioxidant, injectable

composite hydrogel wound dressing to promote the regeneration of infected skin. Journal of colloid and interface science. 2019 Nov 15;556:514-28.

Reply: We now added and cited all the recent works above in the Introduction (on Page 5, second paragraph, citations #24-27), and we compared/listed the benefits of our presented PPN hydrogels over the existing systems (on Page 6, first paragraph). We now also articulated the strengths of our manuscript, including (1) the non-leaching antibacterial and antioxidative dual functions plus the versatility of PPN hydrogel (on Page 24, second paragraph) and (2) the use of 3D DED-HSE model to study the effect of dressing treatment on the re-epithelialization, proliferation and differentiation/maturation of keratinocytes (on Page 23, first and second paragraphs).

Regarding the results, in general the presented work gathers an extensive characterization which demonstrates well the physicochemical properties of the materials and the biofunctionality of the proposed hydrogel and alginate-based fibers while promoting some healing efficacy as wound dressings as compared with the controls, in different in vitro and in vivo models. However, more important than the healing rate is the quality of the regenerated tissue. The characterization of the collagen quality should be assessed to understand the type of new tissue that is being built, besides the information already provided in the histological characterization and presence of wound healing factors.

Reply: Thanks for the positive comments on our extensive characterizations. We will collect the additional data in the revision to characterize the collagen quality.

The hydrogel has very interesting properties and performance, mostly the fact that it has a fast crosslink, and it has good integrity which allows to be periodically changed. However there are no clear evidences on the mechanical properties during the removal of the hydrogel after being in contact with the wound. Does it leave residues? Macroscopic images or a video would be helpful to full demonstrate its integrity. Moreover, the swelling kinetics of hydrogels are not fully discussed in the paper. It is important to better explore the fluid managing capacity of these materials (hydrogel and fibers) over time, according with the type of wound. Water retention and water-vapour permeability is another important property that needs to be assessed to help define the specific type of wound to be addressed. The discussion should clearly bring the publications mentioned above into light.

Reply: Thanks for your positive comments. We will collect these data in the revision, to show the images of dressing removal, swelling kinetics, water retention and water-vapour permeability. Previously, we have also shown the visual appearance of the hydrogels before and after being applied for 2 days of treatment on MRSA USA300-infected wounds of mice (on Figure S5, page S8), in which the morphology of hydrogels remained intact.

For the most part, the work is technically acceptable, but lacks some useful information and a deep interpretation or discussion.

Reply: Thank you for the positive encouragement. We now added the strengths of our manuscript, including the covalently-bound non-leaching antibacterial and antioxidative dual functions plus the versatility of PPN hydrogel (on Page 24, second paragraph – Page 25, first paragraph) and the use of 3D DED-HSE model to study the effect of dressing treatment on the re-epithelialization, proliferation and differentiation/maturation of keratinocytes (on Page 22, third paragraph – Page 23, second paragraph). We will add more useful information and deep interpretation or discussion in the manuscript. (We have also made a draft of this in **Appendix 1** attached.)

Appendix 1 (replies to Reviewer 3 asking us to write more clearly on the following 2 points)

A) Innovation and comparison:

A recent study showed that exosome laden oxygen releasing cryogel OxOBand enhanced collagen deposition, re-epithelialization, neo-vascularization, and reduced oxidative stress to alleviate diabetic wound¹. However, it did not incorporate antibacterial component, as it relied on oxygen release to support the production of ROS, such as H₂O₂ and O²⁻, by the innate host macrophages to fight bacterial infection. A nanocomposite consisting of molybdenum disulfide (MoS₂) nanosheets and cerium dioxide (CeO₂) nanoparticles (NPs) was reported to deliver photothermal antibacterial effect and antioxidative function to treat infected wound². However, this topical ointment left metal components Mo and Ce inside the host body, and moreover it required 808-nm laser treatment to activate the antibacterial effect of MoS₂. A composite hydrogel consisting of (poly)dopamine-modified gelatin, carbon nanotubes (CNTs), and antibiotic doxycycline was reported to have antibacterial and antioxidant properties³. However, the photothermal antibacterial effect of CNT relied on NIR irradiation, and the release of antibiotic from the dressing product posed challenge to spread drug resistance trait. Thermoresponsive hydrogel based on triblock copolymer of caprolactone, glycolide, and ethylene glycol with polydopamine and silver NPs modifications was also reported to exhibit antibacterial and ROS-scavenging properties⁴. Although the hydrogel showed antibacterial effect towards *S. aureus*, Ag NPs are ineffective in killing Gram-negative bacteria such as *P. aeruginosa*, and the in situ sol-gel formation at the wound site also had a chance to leave composition residue in the body.

Our dual-function PPN hydrogel incorporates both strong antibiofilm PIM to kill broad-spectrum MDR bacteria and antioxidant NAC component to alleviate oxidative stress of diabetic wounds. We highlight the synergistic dual functions of contact-active cationic PIM

and antioxidative NAC in a non-leachable wound dressing device to promote healing. Moreover, PPN hydrogel is composed of carbon-based polymers that does not contain antibiotic and metal compound or NP. Unlike many other drug-releasing wound dressings, the sturdy mechanical properties and non-leaching nature of PPN hydrogel allow ease of dressing removal that leaves minimal residue at the wound site. In term of versatility, PPN hydrogel can be fabricated in multiple formats (such as hydrogel film and fiber) that can conform with various shapes of wound, and the fabrication method allows for grafting of different concentrations of active components in order to treat different severities/stages of wounds. These multiple advantages make PPN hydrogel superior over the other wound dressing products.

B) Further Discussion of our results

B.1. Emphasis on non-leaching and combination/dual-functionality non-leaching mechanism:

We highlight the development of PIM with imidazolium main-chain linkers to produce a class of cationic polymer that exhibits superior broad-spectrum antibiofilm activity among other polymers. The novelty of this system is the application of covalently bound antimicrobial PIM and antioxidant NAC that are non-leachable and provide contact-active treatment. Together, the antibiofilm and antioxidative dual functions perform synergistically to accelerate closure of infected diabetic wounds. Four-arm PEG crosslinked with click chemistry forms hydrogel platform with robust mechanical integrity. Such non-leaching and robust dressing does not contaminate the wound or infiltrate into the body and leaves minimal residue at the wound site, which may be categorized under Class II according to the FDA classification (21 CFR 878.4015).

B.2. Emphasis on 3D HSE model:

Wound dressings that cover the site of the wound allow the re-epithelialization. However, full-thickness skin is not generated in this way. Human skin equivalents (HSEs), which have been widely applied as clinical skin replacements and grafts, can also be used as models for drug permeability tests, toxicity screening, or skin injuries and wound treatments. HSEs support remodeling of granulation tissue and formation of scar tissue. In particular, de-epidermised dermis HSE (DED-HSE) is a living *ex vivo* tissue construct in which decellularized dermal scaffolds from human donors are repopulated with allogeneic donor keratinocytes, making it an ideal model to study the keratinocytes transition from proliferating to non-proliferating, differentiated states as they repopulate and heal denuded skin.

In this work, we highlight the use of *ex vivo* 3D DED-HSE model, which is physiologically similar to *in vivo* skin tissue, to study the effects of dressing treatment on the re-epithelialization, proliferation and differentiation potential of keratinocytes, which are important in wound healing. Treatment with our hydrogel formulations showed 39% to 44% closure of wounds after 7 days, whereas silver dressing treatment showed no evidence of closure (Figures 3a-b, S6). We observed a growing wedged-shaped epithelial tongue and greater volume of stratified epithelium in our hydrogel treatment groups, but not in silver dressing groups (Figure 3c). The epidermal thickness data showed that incorporation of PIM(C4) in the hydrogel did not retard re-epithelialization, and incorporation of NAC accelerated re-epithelialization (Table S2). Based on the p63 assay, the presence of PIM(C4) or NAC in the hydrogel did not retard proliferation of keratinocytes. We also found that incorporation of NAC in the hydrogel promote keratinocyte differentiation and maturation, as observed in the K10 and K14 assays. In summary, our PPN hydrogel demonstrated better biocompatibility in reconstructed human skin tissue than silver dressing, and incorporation of NAC in the hydrogel promoted wound healing.

References

1. Shiekh, P. A., Singh, A. & Kumar, A. Exosome laden oxygen releasing antioxidant and antibacterial cryogel wound dressing OxOBand alleviate diabetic and infectious wound healing. *Biomaterials* **249**, 120020 (2020).
2. Ma, T., Zhai, X., Huang, Y., Zhang, M., Zhao, X., Du, Y. & Yan, C. A Smart Nanoplatform with Photothermal Antibacterial Capability and Antioxidant Activity for Chronic Wound Healing. *Advanced Healthcare Materials* **10**, 2100033 (2021).
3. Liang, Y., Zhao, X., Hu, T., Han, Y. & Guo, B. Mussel-inspired, antibacterial, conductive, antioxidant, injectable composite hydrogel wound dressing to promote the regeneration of infected skin. *J. Colloid Interface Sci.* **556**, 514-528 (2019).
4. Ge, P., Chang, S., Wang, T., Zhao, Q., Wang, G. & He, B. An antioxidant and antibacterial polydopamine-modified thermo-sensitive hydrogel dressing for Staphylococcus aureus-infected wound healing. *Nanoscale* (2023).

REVIEWERS' COMMENTS

Reviewer #2 (Remarks to the Author):

The authors have acceptably addressed my comments from the prior review.

I was also asked to look over the responses to Reviewer 1. In my opinion the authors have acceptably addressed Reviewer 1's scientific concerns. The novelty issues highlighted by Reviewer 1 still hold, however, I am under the option that combinatorial strategies and more detailed analyses presented in this study assuage this concern as it is impossible to continue to develop novel biomaterials developments, especially in an established field such as infected dermal wounds.

Reviewer #3 (Remarks to the Author):

The authors have addressed all my comments and suggestions in a clear manner. The authors added an amount of results to complement the work according to the requests. No further comments at this point.